# Prevalence of homebirth preference and associated factors among pregnant women in Ethiopia: Systematic review and meta-analysis

**Jira Wakoya Feyisa**[1]*, **Emiru Merdassa**[1], **Matiyos Lema**[1], **Wase Benti Hailu**[1], **Markos Desalegn**[1], **Adisu Tafari Shama**[1], **Debela Dereje Jaleta**[2], **Gamachis Firdisa Tolasa**[2], **Robera Demissie Berhanu**[3], **Solomon Seyife Alemu**[4], **Sidise Debelo Beyena**[1], **Keno Melkamu Kitila**[5]

1 Department of Public Health, Institute of Health Sciences, Wollega University, Nekemte, Ethiopia, 2 Department of Nursing, College of Health Sciences, Mettu University, Mettu, Ethiopia, 3 Department of Nursing, Institute of Health Sciences, Wollega University, Nekemte, Ethiopia, 4 Department of Midwifery, College of Health Sciences, Madda Walabu University, Shashamane, Ethiopia, 5 Department of Public Health, College of Health Sciences, Mettu University, Mettu, Ethiopia

* jirawakoya462@gmail.com

**Data Availability Statement:** All relevant data are within the paper and its Supporting information files.

## Abstract

### Background

Homebirth preference is the intention/plan to give birth outside health facilities with the help of unskilled birth attendants. The preference to give birth at home without a skilled birth attendant leads to care-seeking delays, intrapartum mortality, multiple stillbirths, and post-partum morbidities and mortality. Therefore, this study aimed to estimate the pooled prevalence of homebirth preference and associated factors among pregnant women in Ethiopia.

### Methods

Search of Google Scholar, Medline, PubMed, Cochrane Library and Web of Science were done for this study from 20th August 2022 to 6th November 2022. For data extraction and analysis, the standardized data extraction checklist and Stata version 14 were used respectively. Sentence as "Cochrane Q test statistics and I2 statistics were used to check heterogeneity of the studies. The pooled prevalence of homebirth preference was estimated using a random-effects model. The association between homebirth preference and independent variables was determined using an odd ratio with a 95% confidence interval. A funnel plot and Egger's test were used to assess publication bias.

### Results

A total of 976 research articles were identified. Seven studies that fulfilled eligibility criteria were included in this systematic review and meta-analysis. The pooled prevalence of home-birth preference in Ethiopia was 39.62% (95% CI 27.98, 51.26). The current meta-analysis revealed that average monthly income <1800 ETB (OR = 2.66, 95% CI 1.44, 4.90) lack of ANC follow-up (OR = 2.57, 95%CI 1.32, 5.01), being multipara (OR = 1.77, 95%CI 1.39, 2.25), poor knowledge about obstetric danger sign (OR = 5.75, 95%CI 1.o2, 32.42), and not

**Funding:** The authors received no specific funding for this work.

**Competing interests:** The authors have declared that no competing interests exist.

**Abbreviations:** ANC, Antenatal care; CI, Confidence interval; EFMOH, Ethiopian Federal Ministry of Health; OR, Odd ratio; SNNPR, Southern Nation nationality and people region; WHO, World Health Organization.

discussing the place of delivery with a partner (OR = 5.89 (95%CI 1.1, 31.63) were significantly associated with homebirth preference.

## Conclusion

This systematic review and meta-analysis examined the substantial prevalence of homebirth preference in Ethiopia which may contribute maternal and child health crisis. The homebirth preference was associated with low average monthly income (<1800 ETB), lack of ANC follow-up, multipara, poor knowledge about obstetric danger signs, and not discussing with their partner the place of delivery. Improving knowledge of pregnant women about the benefit of health facility delivery and obstetric danger signs is necessary to decrease the prevalence of homebirth preference; for these can reduce negative outcomes occurred during delivery.

## Background

Homebirth preference is the intention/plan to give birth outside health facilities with the help of unskilled birth attendants [1]. The preference to give birth at home without a skilled birth attendant leads to care-seeking delays, intrapartum mortality, multiple stillbirths, and postpartum morbidities and mortality [2]. Once a mother plans to give birth at home, the probability of giving birth in healthcare facilities is reduced which increases the risk of maternal fatality rate, but in Sub-Saharan Africa, many women still prefer to give birth at home without a skilled birth attendant [3]. Even though Ethiopia has devised critical strategies, including promoting institutional delivery services for lowering maternal morbidity and mortality, a significant number of women gave birth in their homes in the country [4].

In Ethiopia, a substantial percentage of pregnant mothers prefer to give birth at home [5]. According to the 2019 Mini Ethiopian Demographic and Health Survey report, of all live births in the five years before the survey, only 50% were delivered by a skilled provider [6]. In some parts of Ethiopia regions, more than three-fourths of mothers gave birth at home [4].

Although most maternal health care is free in Ethiopia, home birth rates are reported to be high [7]. Giving birth at home results in premature birth, umbilical cord infection, being accidentally knocked on the head during delivery, breach presentation, and severe maternal anemia which are the main causes of infant mortality [8]. Patients with postpartum hemorrhage who gave birth outside of a hospital had a higher risk of receiving blood transfusion [9].

In Ethiopia, different studies were conducted to determine the homebirth preference and associated factors among pregnant women, which show a great variation across different geographical areas and periods due to different factors such as lack of transportation, low awareness of the advantages of skilled attendance at delivery, little role in making decisions and economic constraints and not discussing the place of delivery with a partner [1, 4, 5, 10].

To guarantee the prevention, detection, and management of problems, all women should have access to skilled care during pregnancy and childbirth. Despite Ethiopia having created several measures for promoting institutional delivery services, home deliveries are still high in different areas [1, 5].

Even though studies have reported on homebirth preference and associated factors among pregnant women in Ethiopia, there is no nationally representative pooled data on homebirth preference and associated factors among pregnant women in Ethiopia. Furthermore, previous studies report varying between regions within the country. Therefore, this systematic review

and meta-analysis aimed to provide pooled representative national data on homebirth preference and associated factors in Ethiopia.

## Methods

### Search strategy

Initially, the PROSPERO database and database of abstracts of reviews of effects (DARE) (http://www.library.UCSF.edu) were searched to check whether published and/or ongoing projects exist related to the topic. The research articles' search strategy, selection of studies, data extraction, and result reporting were done following Preferred Reporting Items for Systematic Reviews and Meta-Analyses (PRISMA) guidelines [11, 12]. A PICO principle was adapted for searching terms. The research articles used for this systematic review and meta-analysis were identified through Google Scholar, Medline/Pub Med, Cochrane Library, the Web of Science, Hinari, Science Direct, ProQuest, African Journals Online, and online university repositories (University of Gondar, Addis Ababa, Jimma, and Haramaya University) search engines by developing search strategies. Boolean operators such as OR, AND, and NOT were used with search terms such as "Preference", "Desire", "need", "intention","choice", "homebirth", "home delivery", "Place of delivery", "place of Birth", "unskilled birth attendant", "associated factors" "determinants", "risk factors", "predictors") "reproductive age women", "pregnant women/mother" and "Ethiopia". Identified research articles were screened to make sure that all relevant literature was included. Literature was downloaded to Endnote (version X7.8) to maintain and manage citations and facilitate the review process [13].

### Eligibility criteria

In this systematic review and meta-analysis, we included:-

1. All studies that were conducted on homebirth preference and/or associated factors among pregnant women in Ethiopia.

2. All types of articles that were published in the form of journal articles

3. Master's theses and dissertations in the English language.

### Outcome measurement

There were two main outcomes. The first outcome of interest was the prevalence, the pooled prevalence of pregnant women preferring homebirth. From primary studies, homebirth preference was operationalized as the choice or desire, or intention of women to give birth at their own homes. The second outcome was the factors associated with homebirth preference, which was determined using the odds ratio (OR) and calculated based on binary outcomes from the included primary studies. The variables included in the review were education, average monthly income, transport access, ANC follow-up (Yes versus No), parity (primipara versus multipara), previous place of delivery (home/health institution), discussion on a place of delivery with a partner (discussed/not discussed) and knowledge (poor/ knowledgeable). Women were asked about the history of ANC follow-up for the current pregnancy. The variable was measured by the binary variable "Yes" for those women who used to visit one and above ANC visits and "No" for women who did not visit ANC during the current pregnancy.

## Data extraction and quality assessment

The Joanna Briggs Institute (JBI) quality appraisal checklist for cross-sectional studies quality assessment instrument was used to rate the included study's level of quality [14]. On Microsoft Excel, three data extractors (JW, EM, and ML) extracted data using a standard data extraction checklist. In the beginning, duplicate articles were removed and search results from databases were combined using reference management software (Endnote version X7.8). Following that, research articles were evaluated and disqualified based on their titles and abstracts. The remaining research articles were evaluated based on full-text publications. The eligibility of the primary studies was assessed following the established inclusion and exclusion criteria. The data extraction checklist for the first outcome (prevalence of homebirth choice) contained the authors' names, year of publication, location (an area where studies were conducted), study design, sample size, response rate, and the number of participants with the homebirth preference. Data were retrieved in two by two table formats for the second outcome (factors associated with homebirth preference), and the log OR was computed based on the results of the source articles. After discussion for potential consensus, discrepancies between the three independent reviewers were resolved by adding more reviewers (MD, AT, WB, and KM). When the included primary articles lacked sufficient information, the corresponding authors of the research articles were contacted via email.

## Data analysis and synthesis

The data were exported to STATA version 14.0 and used to calculate the pooled effect size with 95% CI. To check heterogeneity among the included studies, the Cochran Q test (Chi-squared statistic) and I2 statistic on forest plots were computed. Cochran's Q statistical heterogeneity test is declared statistically significant at $p \leq 0.05$. I2 statistics range from 0 to 100% and I2 statistic values of 0, 25, 50, and 75% were considered as no, low, moderate, and high degrees of heterogeneity, respectively [15]. A random-effects model was used to determine the pooled prevalence of homebirth preference and the pooled effect size of associated factors when a high degree of heterogeneity was observed for the first and second outcomes. To identify the source of potential random variation, subgroup analysis was conducted based on the region where the studies were conducted and based on the timing of the study. Meta-regression was computed to see the presence of statistically significant heterogeneity. Publication bias was assessed by using a funnel plot. The symmetry of the funnel plot is an indicator of the absence of publication bias (Fig 5). In addition, Egger's (p-value = 0.079 weighted regression and Begg's tests (p-value = 0.051) were used to check the absence of publication bias. Statistical nonsignificance of publication bias was declared at a p-value of greater than 0.05 [16].

## Result

976 studies were found via the data bases, then 646 articles recorded from these studies were found to be duplicates and were eliminated while 303 irrelevant research publications were excluded from our analysis after being reviewed for titles and abstracts. Twenty studies were eliminated after reviewing the whole texts of the remaining publications because they did not adhere to the predetermined eligibility requirements. The final systematic review and meta-analysis included the remaining seven (7) research articles (Fig 1).

## Characteristics of included studies

All of the seven studies included in this study were published between 2019 to 2022 in peer review journals [5, 10, 17–21]. A total of 3458 study participants were included in the current

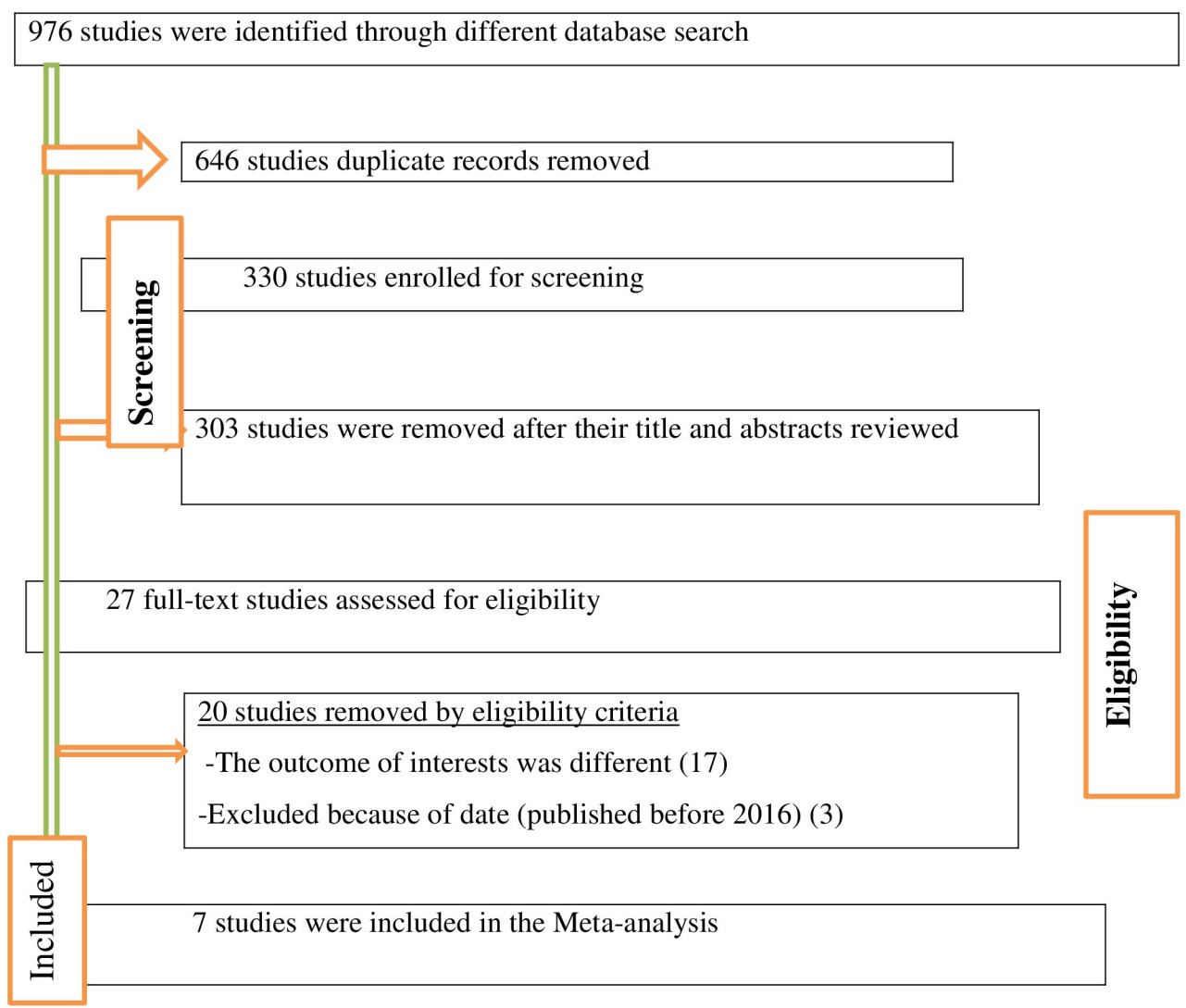

**Fig 1. PRISMA flow diagram of included studies in the systematic review and meta-analysis of the prevalence of homebirth preference and associated factors among pregnant women in Ethiopia, 2022.**

**Table 1. Summary of included studies on the prevalence of homebirth preference in Ethiopia, 2022.**

| SN | Author | Publication year | Region | Study area | Study design | Sample size | Quality score | Prevalence(95%CI) |
|---|---|---|---|---|---|---|---|---|
| 1 | Alemu et al. [10] | 2019 | SNNR | Wonago | Cross-sectional | 769 | 7 | 25.62(22.53,28.70) |
| 2 | Kahsay et al. [17] | 2019 | Afar | Zone 3 | Cross-sectional | 357 | 7 | 69.75(64.98,74.51) |
| 3 | Mekie and Taklual [18] | 2019 | Amhara | Simada | Cross-sectional | 346 | 8 | 56.36(51.13,61.58) |
| 4 | Tsegaye et al. [19] | 2019 | Amhara | Debretabor | Cross-sectional | 394 | 7 | 29.19(24.70,33.68) |
| 5 | Bekuma et al. [20] | 2020 | Oromia | Jimma Arjo | Cross-sectional | 506 | 8 | 39.53(35.27,43.79) |
| 6 | Alemu et al. [21] | 2022 | SNNPR | Arba Minch S.S | Cross-sectional | 408 | 9 | 24.02(19.87,28.17) |
| 7 | Teferi et al. [5] | 2022 | PMA | PMA survey | Cross-sectional | 678 | 9 | 33.33(29.78,36.88) |

Note: CI; confidence interval, SNNPR; Southern Nation Nationalities, and people region

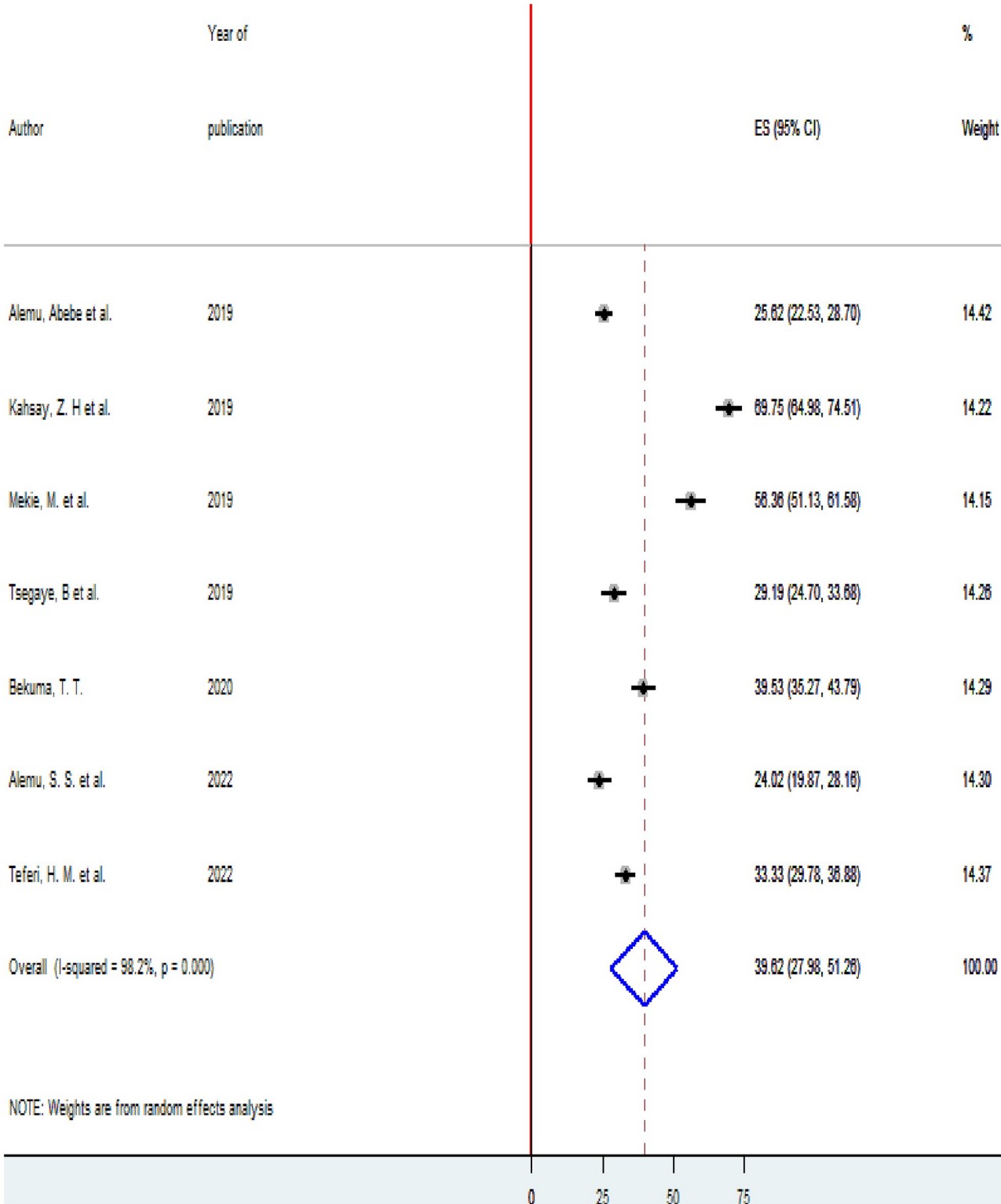

**Fig 2. Forest plot of the pooled estimate of the prevalence of homebirth preference in Ethiopia, 2022.**

**Table 2. Meta-regression analysis based on sample size and year of publication.**

| Variables | Coefficients | p-value |
|---|---|---|
| Year of publication | -4.295715 | 0.421 |
| Sample size | -.0471067 | 0.309 |

systematic review and meta-analysis. The smallest sample size was 346 from a study conducted in the Amhara region, Simada district [18], and the largest sample size was 769 from a study conducted in SNNPR [10]. All included research articles were cross-sectional studies with 4 prospective cross-sectional study designs nature [5, 17, 19, 21] and 3 retrospective cross-sectional study designs [10, 18, 20]. Regarding study setting, two studies were conducted in the Amhara region [18, 19], two studies in the SNNPR [10, 21], one study was from the Afar region [17], one study was conducted in the Oromia region [20] and one study conducted from PMA survey data [5] (Table 1).

## Prevalence of homebirth preference

High heterogeneity was observed across the included studies (I2 = 98.2, p < 0.001) thus, a random-effects model was used to estimate the pooled prevalence of homebirth Preference. The prevalence of homebirth Preference was 39.62% (95% CI 27.98, 51.26). The highest 69.75(95% CI 64.98,74.51) prevalence of homebirth Preference was observed in Zone 3, Afar region [17] and the lowest 24.02(19.87,28.17) prevalence of homebirth Preference was reported in Arba Minch zuria surveillance site, SNNPR [21] (Fig 2).

To check for underlying heterogeneity, meta-regression models were done by using sample size and year of publication, but there was statistically insignificant underlying heterogeneity (p = 0.309) and (p = 0.421), respectively (Table 2).

## Subgroup analysis

To see heterogeneity among the included studies, subgroup analysis was executed based on the area where the studies were conducted and the timing of studies (prospective and retrospective cross-sectional studies). According to where the studies conducted, the highest prevalence of homebirth Preference 47.49(95% CI 26.59,68.39) was observed in other (Oromia, Afar, and PMA survey data regions) [5, 17, 20] and the lowest prevalence of homebirth Preference 25.05 (22.57,27.52) was reported in SNNPR (Fig 3). According to the timing of the study, nearly the prevalence of homebirth Preference was equal among retrospective studies 40.4% (95% CI 23.28, 57.52) and prospective studies 39.04 (95% CI 20.47, 57.81) (Fig 4).

## Publication bias

To see the presence of publication bias, the graphical funnel plot and Egger's test at a 5% significance level were executed. The visual examination of the funnel plot presented symmetrically which is an indicator for the absence of publication bias (Fig 5). Egger's test also showed the absence of publication bias at a 5% significance level (p = 0.079).

## Sensitivity analysis

Outliers were detected by Sensitivity analysis showing that there was no single study that influence the overall included studies (Fig 6).

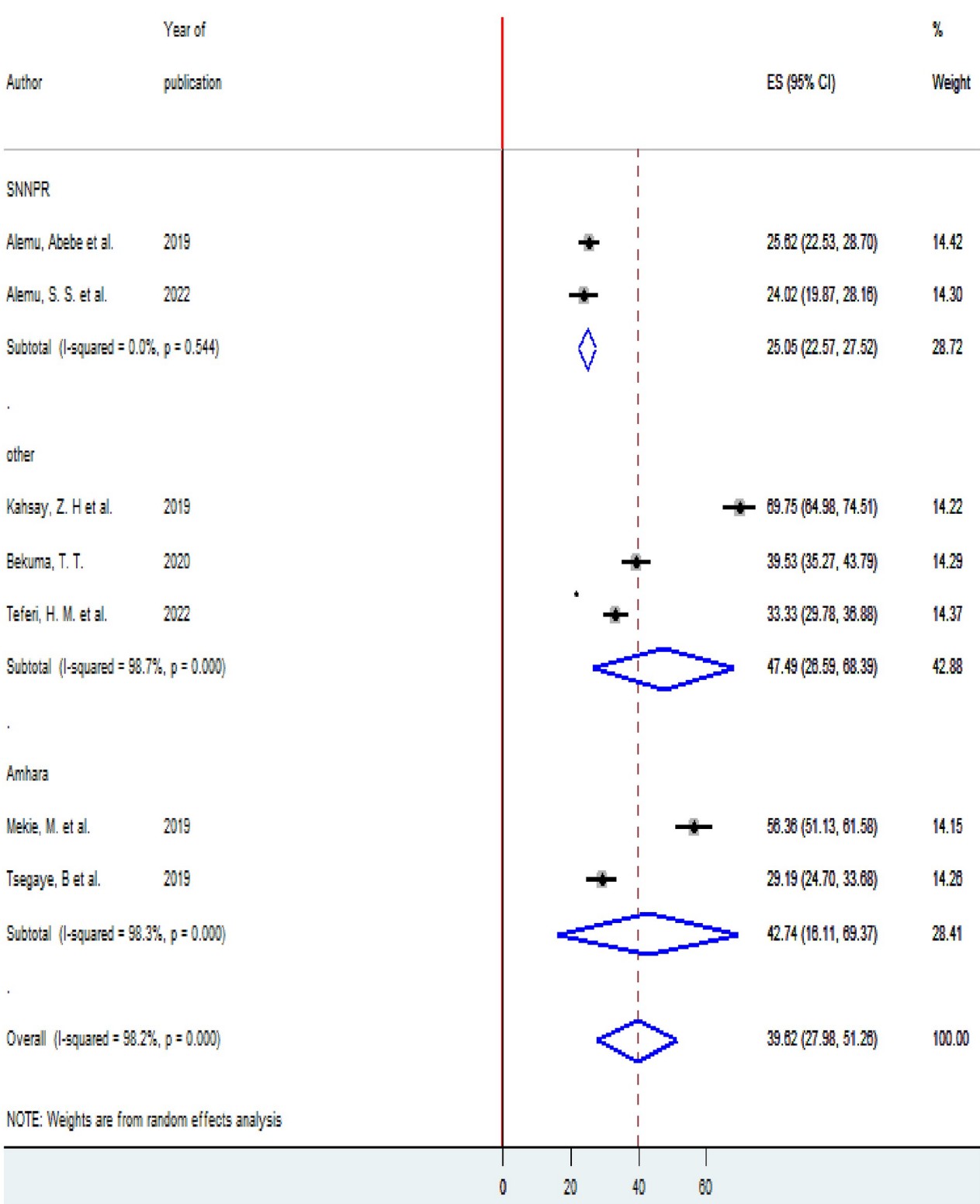

**Fig 3. Subgroup analysis of the prevalence of homebirth preference among pregnant women in Ethiopia based on the region, 2022.**

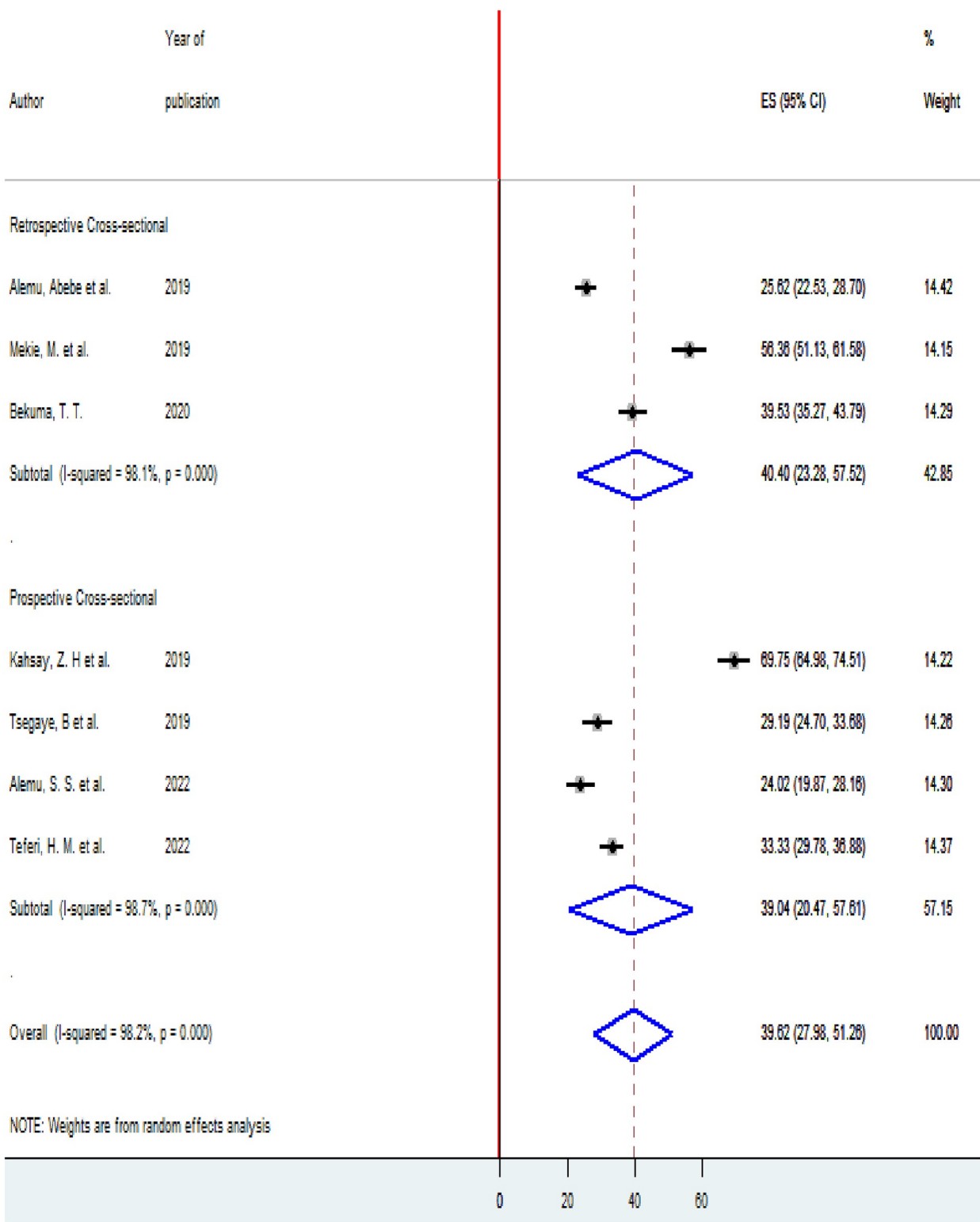

**Fig 4. Subgroup analysis of the prevalence of homebirth preference among pregnant women in Ethiopia based on the timing of the study, 2022.**

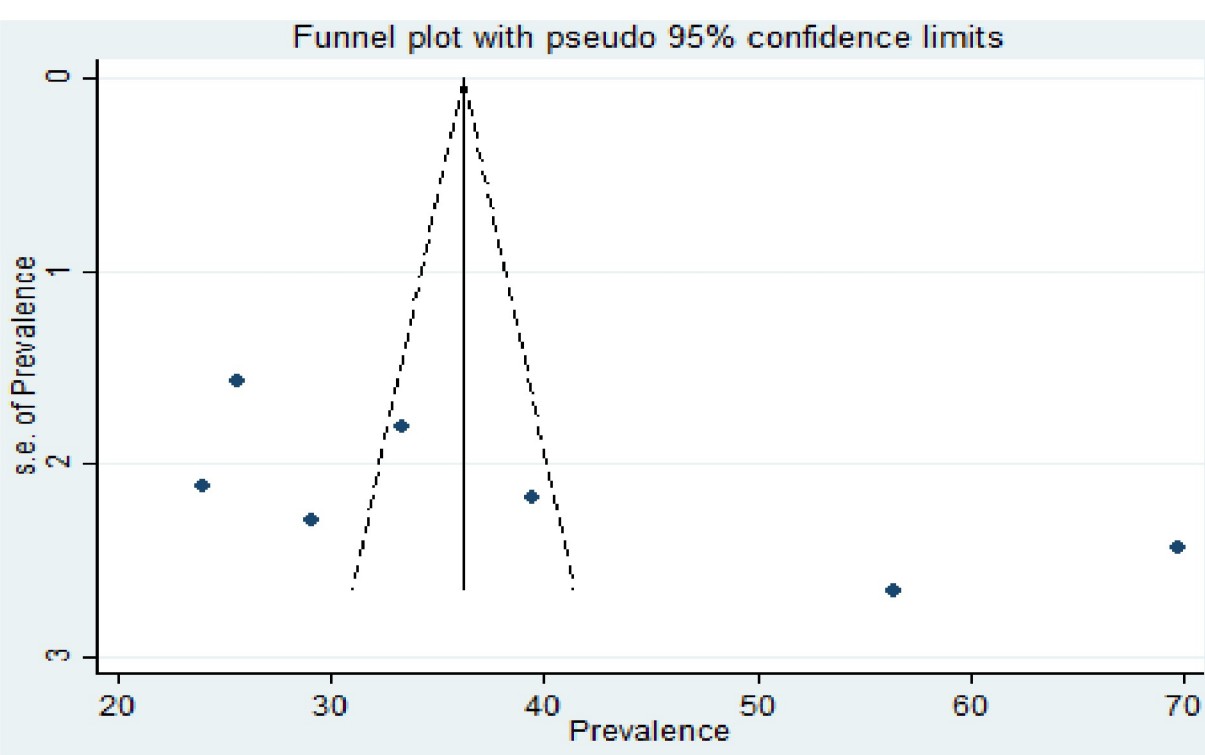

**Fig 5. Funnel plot with 95% confidence limit of the prevalence of homebirth preference in Ethiopia, 2022.**

## Factors associated with homebirth preference

### Association between education and homebirth preference

To examine the association between maternal Education and homebirth preference, three studies were included in the analysis [10, 19, 20]. The pooled association between maternal education and homebirth preference was estimated by a random-effects model (I2 = 85.3%, P-value = 0.001). The pooled result of the analysis indicates that the association between maternal education and homebirth preference was not statistically significant (OR = 0.72, CI 0.43, 1.19) (Fig 7).

### Association between monthly income and homebirth preference

To examine the association between average monthly income and homebirth preference, two studies were included in the analysis [18, 19]. The pooled association between average monthly income and homebirth preference was estimated by using A random-effects model due to moderate heterogeneity (I2 = 69.0%). The pooled result of the analysis indicated that average monthly income < 1800 ETB was significantly associated with homebirth preference. Mothers who got <1800 ETB prefer homebirth 2.66 times more likely as compared to those who get 1800 ETB and above (OR = 2.66, 95% CI 1.44, 4.90) (Fig 8).

### Association between ANC follow up and homebirth preference

Five studies [5, 10, 18–20] were analyzed to look at the association between ANC follow-up and desire for homebirth. Due to substantial heterogeneity (I2 = 88.1%, p-value 0.001), a random-effects model was employed to quantify the pooled association between ANC follow-up

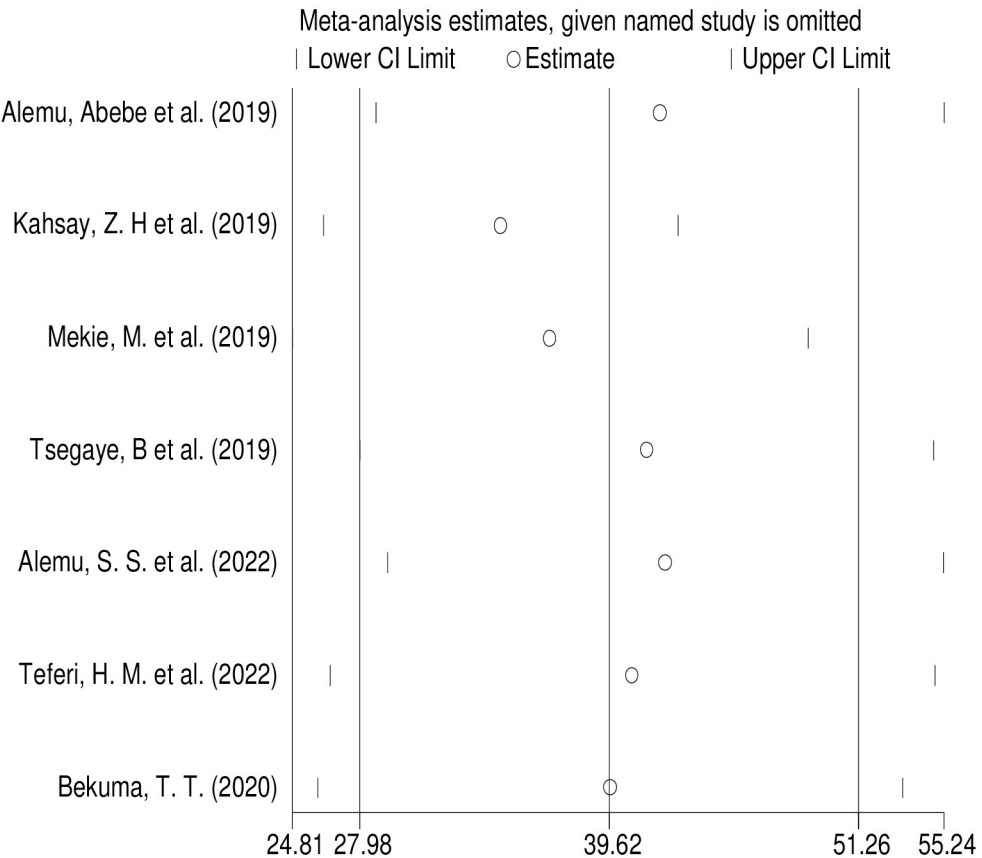

**Fig 6. Result of sensitivity analysis of homebirth preference in Ethiopia, 2022.**

and preference for home birth. The analysis's combined findings show that choosing to give delivery at home was associated with not receiving ANC follow-up. Women who did not have ANC follow-up during pregnancy were 2.57 times more likely to desire homebirth than those who did (OR = 2.57, 95%CI 1.32, 5.01) (Fig 9).

## Association between parity and homebirth preference

To examine the association between parity and homebirth preference, two studies were included in the analysis [5, 10]. A fixed-effect model was employed to estimate the pooled association between parity and homebirth preference because of no heterogeneity (I2 = 0%, p-value < 0.576). The pooled result of the analysis indicates that parity was significantly associated with homebirth preference. Women of multipara prefer homebirth 1.77 times more likely as compared to those primiparas (OR = 1.77, 95%CI 1.39, 2.25) (Fig 10).

## Association between knowledge of obstetric danger signs and homebirth preference

Two studies [10, 21] were analyzed to determine whether knowledge of obstetric danger signs and choice of homebirth were related. Due to substantial heterogeneity (I2 = 95.2%, p-value 0.001), to quantify the pooled association between knowledge of obstetric danger signs and preference for homebirth a random-effects model was employed. Poor knowledge was

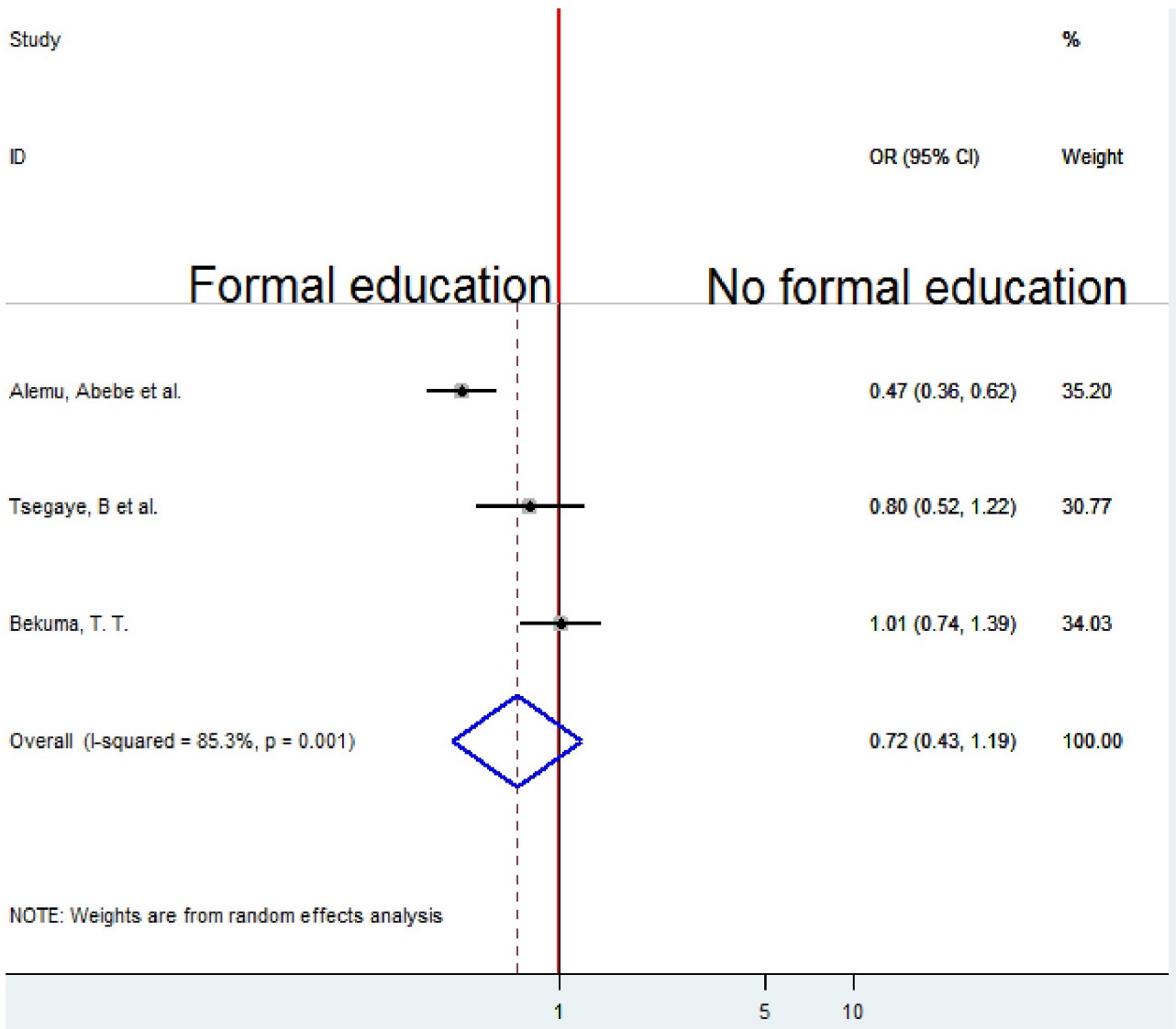

**Fig 7. Forest plot of the pooled estimate of the association between education and homebirth preference in Ethiopia, 2022.**

significantly associated with a desire for home birth, according to the analysis's pooled result. When compared to women who were aware of obstetric hazard indications, those with poor knowledge were 5.75 times more likely to prefer homebirth (OR = 5.75, 95%CI 1.o2, 32.42) (Fig 11).

## Association between transportation service accessibility and homebirth preference

To examine the association between transportation service access and homebirth preference, three studies were included in the analysis [18, 20, 21]. To estimate the pooled association between transportation service access and homebirth preference A random-effects model was employed because of significant heterogeneity (I2 = 96.9%, p-value < 0.001). The pooled result

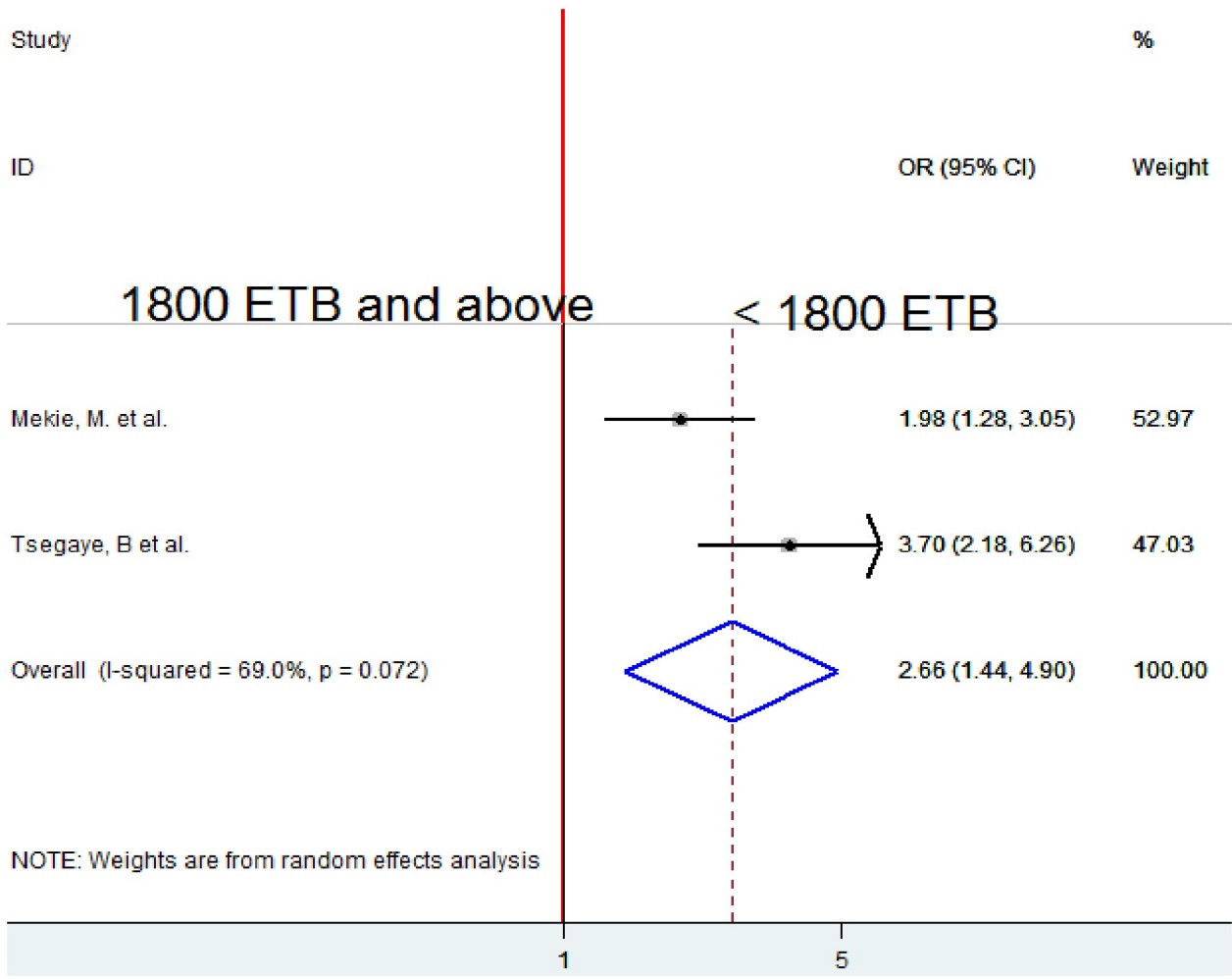

**Fig 8. Forest plot of the pooled estimate of the association between monthly income and homebirth preference in Ethiopia, 2022.**

of the analysis indicated an insignificant association between transportation service access and homebirth preference (Fig 12).

## Association between delivery place discussion and homebirth preference

To examine the association between delivery place discussion and homebirth preference, two studies were included in the analysis [5, 21]. The pooled association between delivery place discussion and homebirth preference was examined by random-effects model, for high heterogeneity (I2 = 95.6%, p-value < 0.001). The pooled result of the analysis indicates that not having a delivery place discussion with a partner was significantly associated with homebirth preference. Women who did not discuss with their partners a place of delivery prefer homebirth 5.89 times more likely as compared to those who discussed with their partners (OR = 5.89 (95%CI 1.1, 31.63) (Fig 13).

## Association between the place of last delivery and homebirth preference

To examine the association between the place of last delivery and homebirth preference, two studies were included in the analysis [10, 18]. The pooled result of the association between the

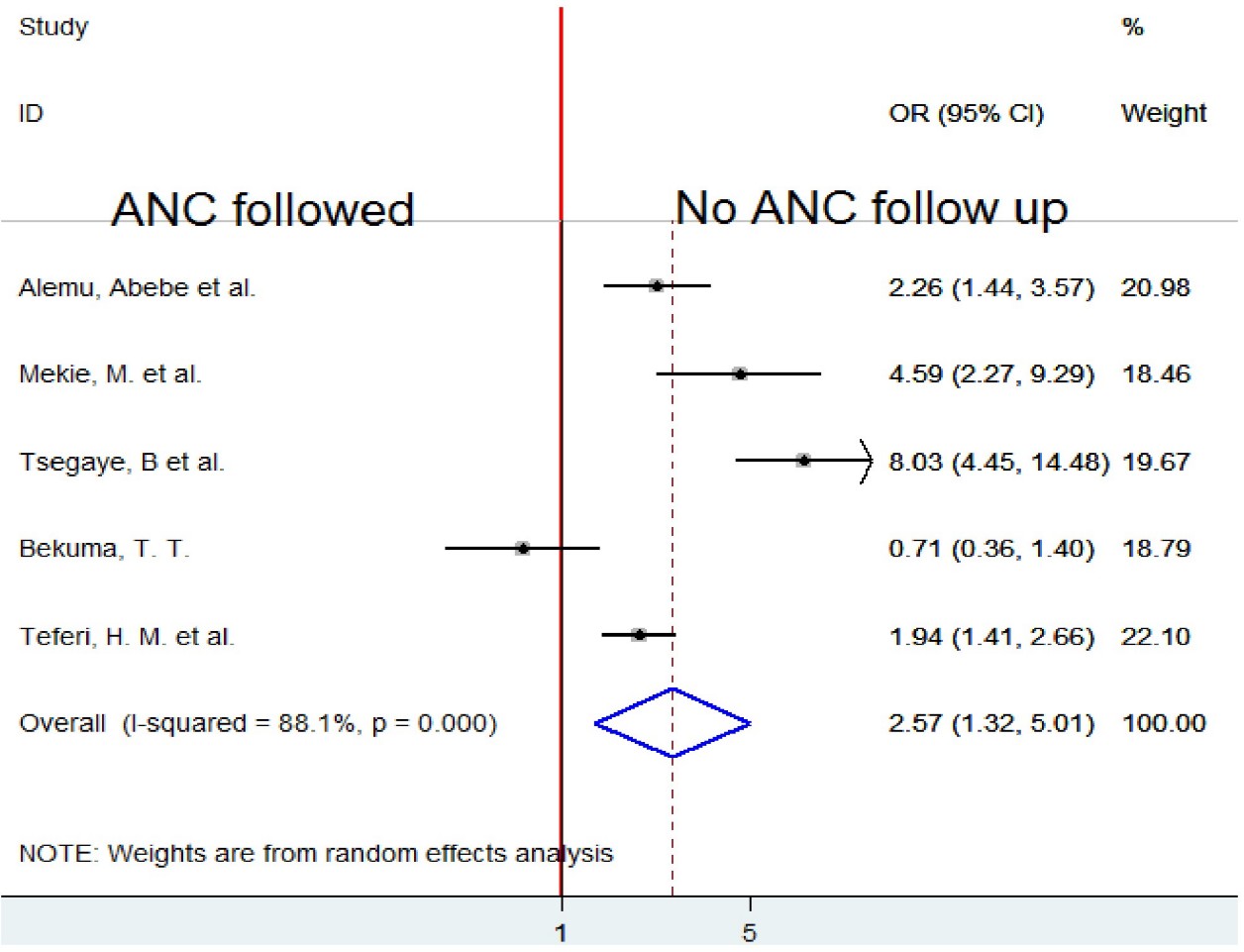

**Fig 9. Forest plot of the pooled estimate of the association between ANC follow-up and homebirth preference in Ethiopia, 2022.**

place of last delivery and homebirth preference was assessed by a random-effects model for the reason of high heterogeneity (I2 = 97.2%, p-value < 0.001). The pooled result of the analysis indicated an insignificant association between the place of last delivery and homebirth preference (Fig 14).

## Discussion

This systematic review and meta-analysis assessed homebirth preference and associated factors in Ethiopia, 2022. The prevalence of homebirth preference among pregnant women in Ethiopia was 39.62% (95% CI 27.98, 51.26). Even though WHO and EFMOH inspire every woman to give birth at a health institution, this systematic review and meta-analysis revealed that the prevalence of homebirth preference is high. This result is in line with a study done in Nigeria 39.3% [22]. But the finding of this systematic review and meta-analysis is higher than the finding of the study conducted in Tanzania (25.5%) [23]. A possible explanation for this inconsistent finding might be due to the local versus national study populations involved in these studies. The discrepancy could be due to the difference in the study subjects. The previous study was undertaken among all reproductive women whereas most of the participants of this systematic review and meta-analysis were pregnant mothers. Pregnant women's favorite for

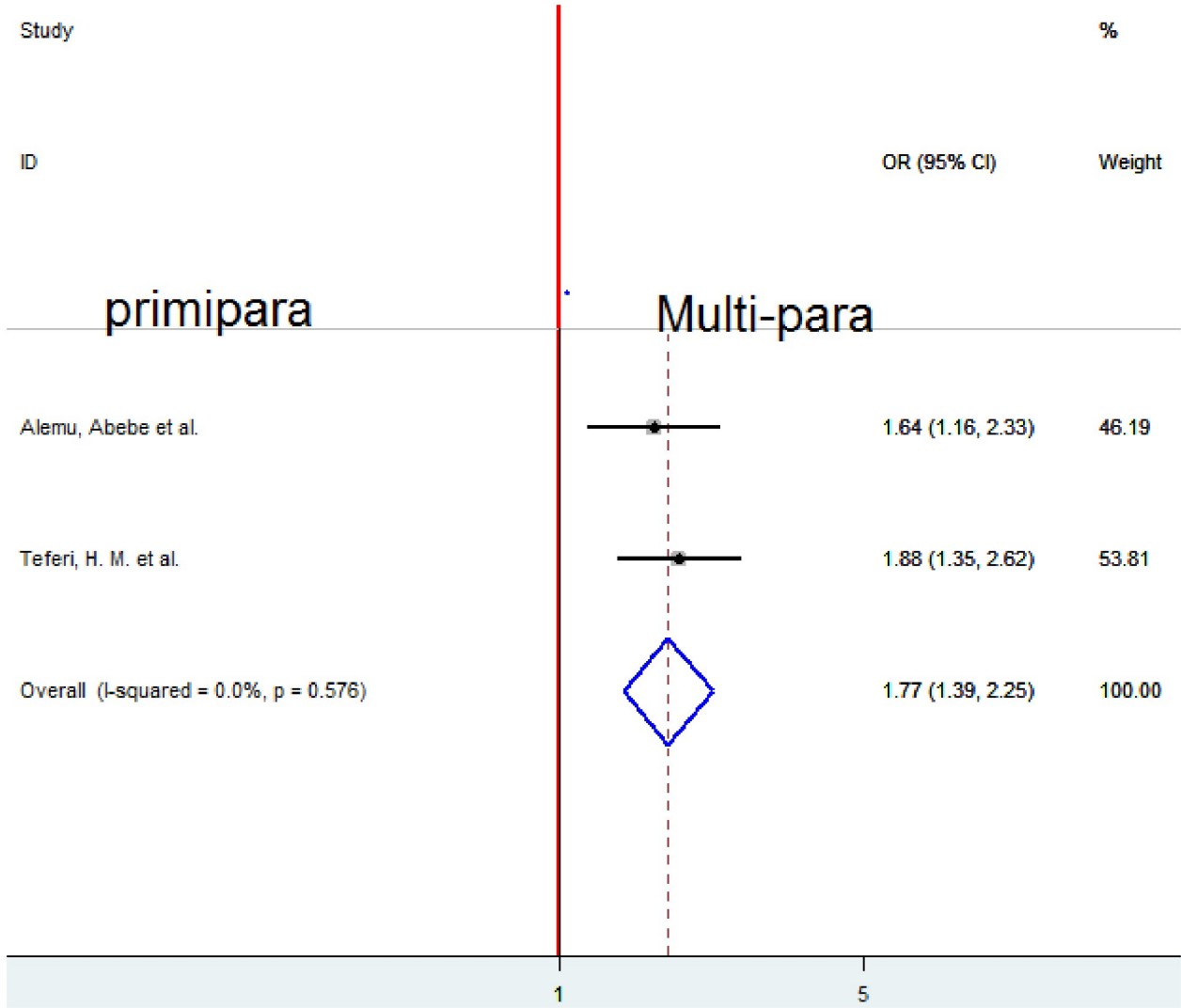

**Fig 10. Forest plot of the pooled estimate of the association between parity and homebirth preference in Ethiopia, 2022.**

homebirth declines as gestational age upturns because most pregnant women start ANC follow-up after the second trimester. Therefore, they get counseling about the advantages of institutional delivery.

This systematic review and meta-analysis revealed that low average monthly income (< 1800 ETB) was significantly associated with homebirth preference. Women with low average monthly income were about 3 fold times more likely to prefer homebirth than their counterparts. This finding is consistent with the study done in Tanzania [24]. Health care in Ethiopia for pregnant women is free. The reason might be healthcare facilities seldom run out of stock, demanding pregnant mothers charge some expenses for supplies or services. Besides, women from a distance pay a transportation fee to go to a health facility for maternity care. Therefore, pregnant women with better incomes are better to pay for transport and other health care payments.

Poor knowledge about obstetrical danger signs was significantly associated with home birth preference. Homebirth preference among women who had poor knowledge about

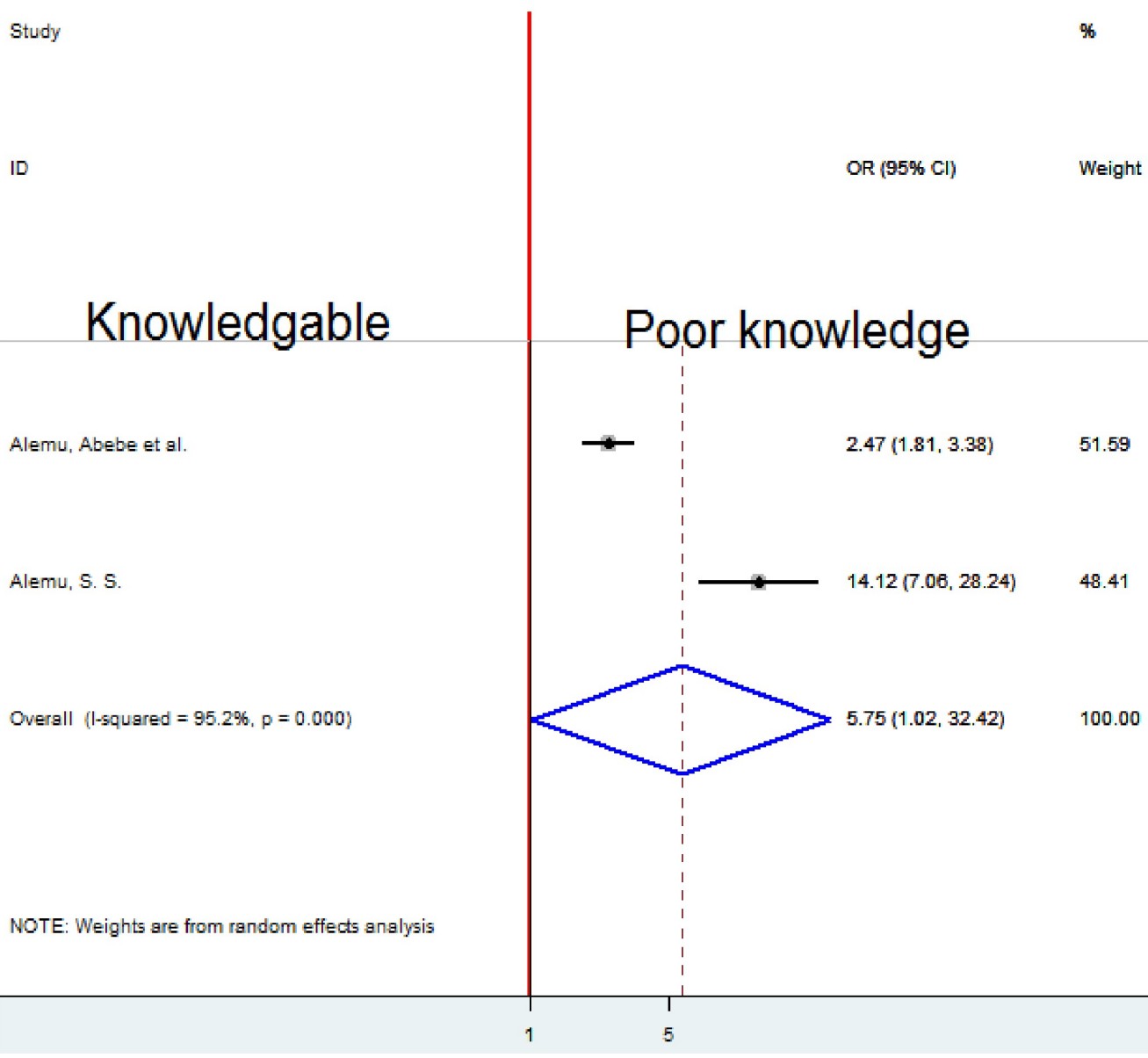

**Fig 11. Forest plot of the pooled estimate of association knowledge about danger signs and homebirth preference in Ethiopia, 2022.**

danger signs was about 6 fold times higher compared to knowledgeable women. This is supported by the findings of studies undertaken in Ghana [25]. The possible explanation might be that maternal education and counseling which are given during ANC follow-up can change and boost the knowledge of pregnant mothers. As a result, this systematic review and meta-analysis indicated failure to attend ANC was significantly associated with homebirth preference. This result reveals that someone who has ANC follow-up can have advice and counseling regarding obstetric danger signs. This can enhance women's intention to give birth at the health institution. As the knowledge of women about obstetric danger signs is boosted, understanding the benefits of health institution delivery becomes more efficient than homebirth.

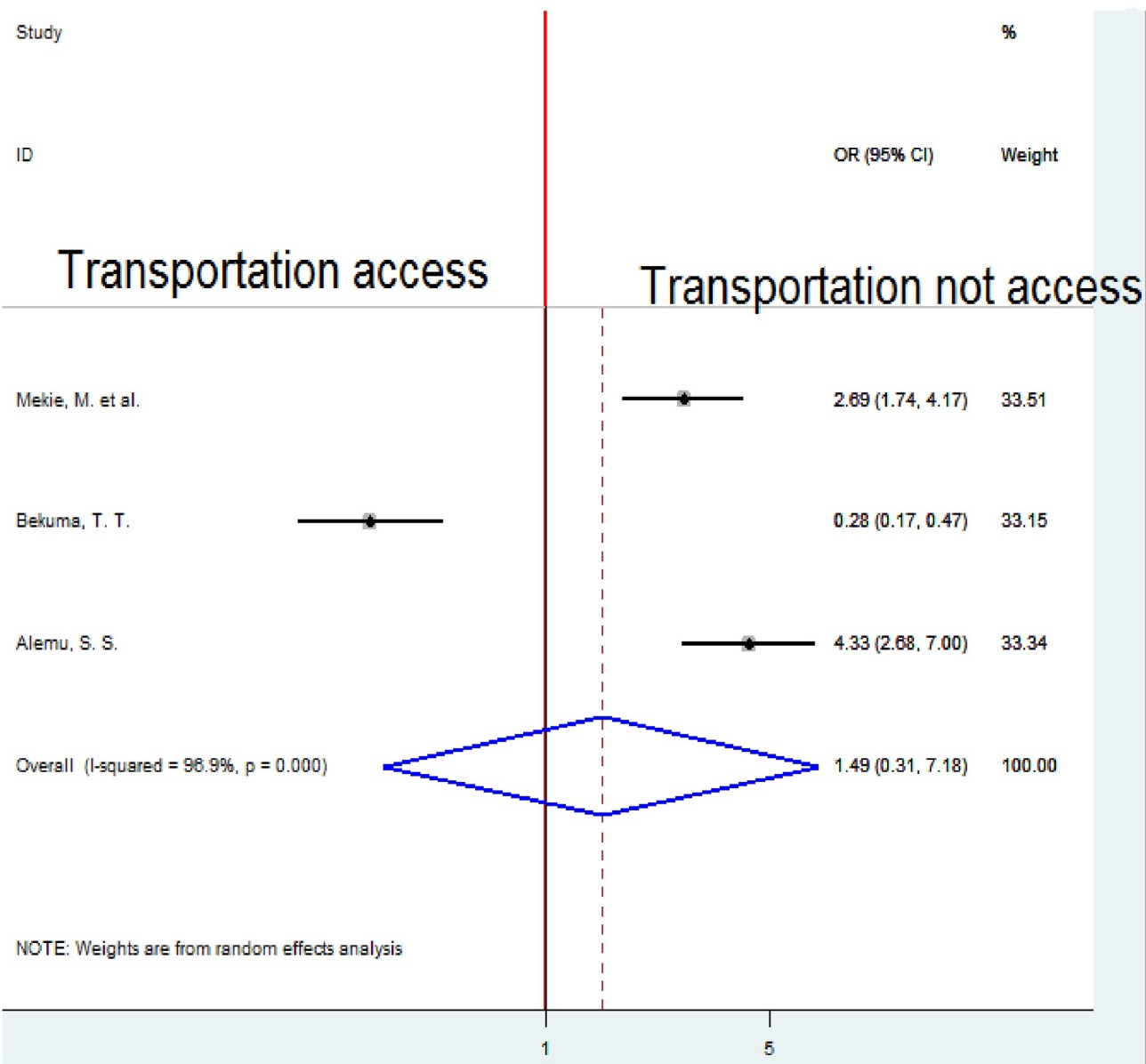

**Fig 12. Forest plot of the pooled estimate of the association between transportation access and homebirth preference in Ethiopia, 2022.**

In this systematic review and meta-analysis, multipara mothers were about 2 fold times more likely to prefer homebirth compared to nulliparous women in Ethiopia. This finding is in line with a study conducted in Ghana [26]. This might be due to their previous childbirth understanding.

Mothers not attended ANC visits in the current pregnancy were about 3 fold times more likely to prefer home delivery than those who attended ANC in Ethiopia. This finding is consistent with studies conducted in Nigeria [22]. It might be women who had ANC visits by health professionals had better opportunities to be counseled regarding birth preparedness, complication readiness, and place of delivery which may enhance delivery at a health institution than preferring to give birth at their own home.

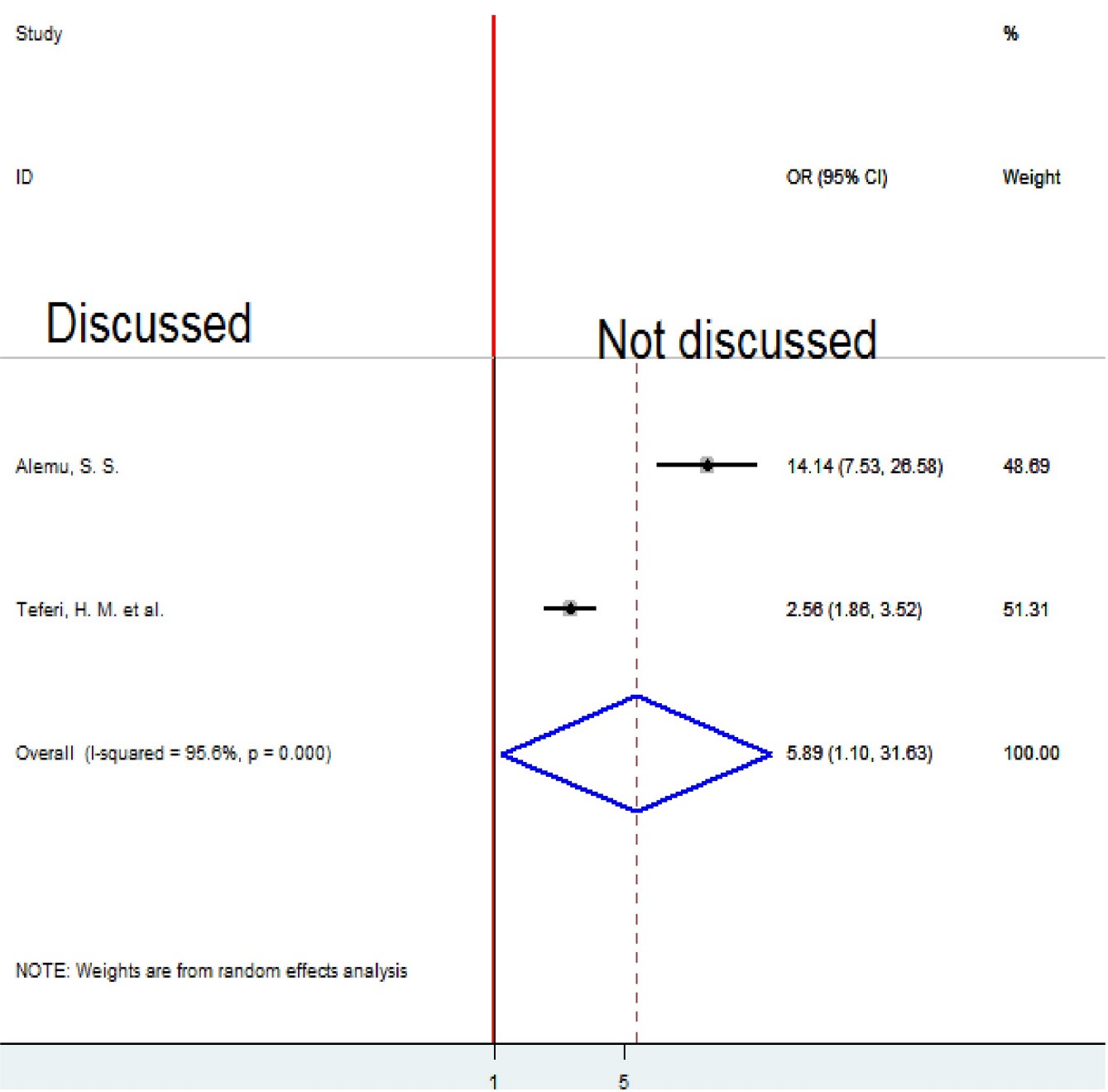

**Fig 13. Forest plot of the pooled estimate of the association between delivery place discussion and homebirth preference in Ethiopia, 2022.**

This systematic review and meta-analysis found that women who did not discuss the place of their delivery with their partners were about six times more likely to desire a homebirth than those who did. Studies conducted in Tanzania and Mozambique in the past have produced consistent results [27]. If a woman who is pregnant is financially reliant on her spouse and the partner is unaware of the benefits of institutional birth, he may be able to choose the location of the delivery on his own because he is the source of the money. When couples debate where to give birth, there is a chance that the advantages and disadvantages of

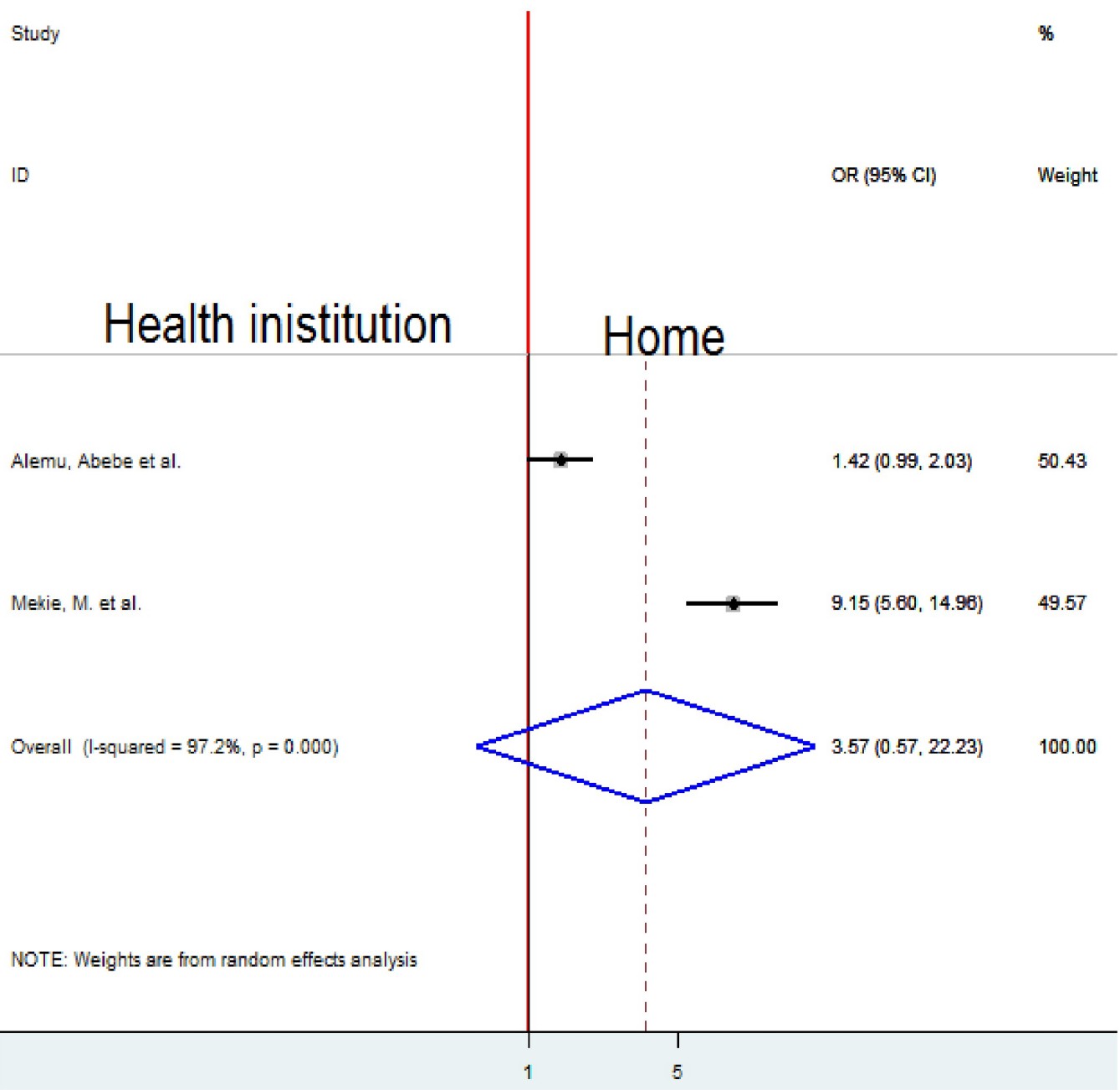

**Fig 14. Forest plot of the pooled estimate of the association between place of last delivery and homebirth preference in Ethiopia, 2022.**

homebirth will come up. Partners disclose that the burden might come to them due to homebirth. They may decide to give birth at a health institution. Thus, why, not discussing with a partner may result in home birth.

## Strength and limitation

This systematic review and meta-analysis has strengths; various databases were used to search for literature, both published and unpublished studies were searched and relevant studies were

included after intensive quality assessment. Because less than 10 studies were included in the final analysis, it is difficult to get the exact estimation of publication bias from the funnel plot.

## Conclusion

This systematic review and meta-analysis examined the substantial prevalence of homebirth preference in Ethiopia which may contribute maternal and child health crisis. The homebirth preference was associated with a low average monthly income (<1800 ETB), lack of ANC follow-up, multipara, poor knowledge about obstetric danger signs, and not discussing with their partner a place of delivery. Furthermore, a discrepancy was also found in different regions of the country. So, community-based awareness creation is mandatory to improve ANC follow-up and enhance knowledge about obstetric danger signs. The health policy of the country should include a mechanism that encourages couples to discuss together the exact choice of the right place of delivery to minimize perinatal and postpartum complications. Community health workers in the country should routinely deliver the basic knowledge of obstetric danger signs to pregnant mothers at a community level.

## Supporting information

**S1 Checklist. PRISMA 2020 checklist.**
(DOCX)

**S1 Data.**
(ZIP)

## Acknowledgments

We would like to thank all authors of the studies included in this systematic review and meta-analysis.

## Author Contributions

**Conceptualization:** Jira Wakoya Feyisa, Emiru Merdassa, Matiyos Lema, Wase Benti Hailu, Markos Desalegn, Adisu Tafari Shama, Debela Dereje Jaleta, Gamachis Firdisa Tolasa, Robera Demissie Berhanu, Solomon Seyife Alemu, Sidise Debelo Beyena, Keno Melkamu Kitila.

**Data curation:** Jira Wakoya Feyisa, Emiru Merdassa, Matiyos Lema, Wase Benti Hailu, Markos Desalegn, Adisu Tafari Shama, Debela Dereje Jaleta, Gamachis Firdisa Tolasa, Robera Demissie Berhanu, Solomon Seyife Alemu, Sidise Debelo Beyena, Keno Melkamu Kitila.

**Formal analysis:** Jira Wakoya Feyisa, Emiru Merdassa, Matiyos Lema, Wase Benti Hailu, Markos Desalegn, Adisu Tafari Shama, Debela Dereje Jaleta, Gamachis Firdisa Tolasa, Robera Demissie Berhanu, Solomon Seyife Alemu, Sidise Debelo Beyena, Keno Melkamu Kitila.

**Funding acquisition:** Emiru Merdassa, Wase Benti Hailu, Markos Desalegn, Debela Dereje Jaleta.

**Investigation:** Jira Wakoya Feyisa, Emiru Merdassa, Matiyos Lema, Wase Benti Hailu, Markos Desalegn, Adisu Tafari Shama, Debela Dereje Jaleta, Gamachis Firdisa Tolasa, Robera Demissie Berhanu, Solomon Seyife Alemu, Sidise Debelo Beyena, Keno Melkamu Kitila.

**Methodology:** Jira Wakoya Feyisa, Emiru Merdassa, Wase Benti Hailu, Markos Desalegn, Adisu Tafari Shama, Debela Dereje Jaleta, Gamachis Firdisa Tolasa, Robera Demissie Berhanu, Solomon Seyife Alemu, Sidise Debelo Beyena, Keno Melkamu Kitila.

**Project administration:** Jira Wakoya Feyisa, Emiru Merdassa, Matiyos Lema, Markos Desalegn, Adisu Tafari Shama, Debela Dereje Jaleta, Gamachis Firdisa Tolasa, Sidise Debelo Beyena, Keno Melkamu Kitila.

**Resources:** Jira Wakoya Feyisa, Emiru Merdassa, Matiyos Lema, Markos Desalegn, Adisu Tafari Shama, Debela Dereje Jaleta, Gamachis Firdisa Tolasa, Robera Demissie Berhanu, Solomon Seyife Alemu, Keno Melkamu Kitila.

**Software:** Jira Wakoya Feyisa, Emiru Merdassa, Wase Benti Hailu, Markos Desalegn, Adisu Tafari Shama, Debela Dereje Jaleta, Gamachis Firdisa Tolasa, Solomon Seyife Alemu, Sidise Debelo Beyena, Keno Melkamu Kitila.

**Supervision:** Jira Wakoya Feyisa, Emiru Merdassa, Matiyos Lema, Wase Benti Hailu, Markos Desalegn, Adisu Tafari Shama, Debela Dereje Jaleta, Gamachis Firdisa Tolasa, Robera Demissie Berhanu, Solomon Seyife Alemu, Sidise Debelo Beyena, Keno Melkamu Kitila.

**Validation:** Jira Wakoya Feyisa, Emiru Merdassa, Wase Benti Hailu, Markos Desalegn, Adisu Tafari Shama, Debela Dereje Jaleta, Gamachis Firdisa Tolasa, Robera Demissie Berhanu, Solomon Seyife Alemu, Sidise Debelo Beyena, Keno Melkamu Kitila.

**Visualization:** Jira Wakoya Feyisa, Emiru Merdassa, Wase Benti Hailu, Markos Desalegn, Adisu Tafari Shama, Debela Dereje Jaleta, Gamachis Firdisa Tolasa, Robera Demissie Berhanu, Solomon Seyife Alemu, Sidise Debelo Beyena, Keno Melkamu Kitila.

**Writing – original draft:** Jira Wakoya Feyisa, Emiru Merdassa, Matiyos Lema, Wase Benti Hailu, Markos Desalegn, Adisu Tafari Shama, Debela Dereje Jaleta, Gamachis Firdisa Tolasa, Robera Demissie Berhanu, Solomon Seyife Alemu, Sidise Debelo Beyena, Keno Melkamu Kitila.

**Writing – review & editing:** Jira Wakoya Feyisa, Emiru Merdassa, Matiyos Lema, Wase Benti Hailu, Markos Desalegn, Adisu Tafari Shama, Debela Dereje Jaleta, Gamachis Firdisa Tolasa, Robera Demissie Berhanu, Solomon Seyife Alemu, Sidise Debelo Beyena, Keno Melkamu Kitila.

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
