## [Decision Letter · Decision Letter 0]

20 Mar 2023

PONE-D-23-02877PREVALENCE OF HOMEBIRTH PREFERENCE AND ASSOCIATED FACTORSAMONG PREGNANT WOMEN IN ETHIOPIA: SYSTEMATIC REVIEW AND META-ANALYSISPLOS ONE

Dear Dr. Feyisa,

Thank you for submitting your manuscript to PLOS ONE. After careful consideration, we feel that it has merit but does not fully meet PLOS ONE’s publication criteria as it currently stands. Therefore, we invite you to submit a revised version of the manuscript that addresses the points raised during the review process.

General comments:

I would like to thank the authors for coming with such an important manuscript. However, there are a lot of issues that needs to be improved. Please address the following issues:

1. The language of the manuscript needs to be improved for publication.

2. The problem studied is not well explained in a focused manner?

3. I am not convinced why only studies published after 2016 are included? I recommend you to include those 3 studies published before 2016.

4. Please revise the discussion section

Moreover, please carefully address all reviewers comments included here.

We look forward to receiving your revised manuscript.

Kind regards,

Biruk Bogale Wolde

Academic Editor

PLOS ONE

Journal Requirements:

2. Thank you for submitting the above manuscript to PLOS ONE. During our internal evaluation of the manuscript, we found significant text overlap between your submission and previous work in the Abstract, Methods, Results and Discussion section. We would like to make you aware that copying extracts from previous publications, especially outside the methods section, word-for-word is unacceptable. In addition, the reproduction of text from published reports has implications for the copyright that may apply to the publications. Please revise the manuscript to rephrase the duplicated text, cite your sources, and provide details as to how the current manuscript advances on previous work. Please note that further consideration is dependent on the submission of a manuscript that addresses these concerns about the overlap in text with published work. We will carefully review your manuscript upon resubmission and further consideration of the manuscript is dependent on the text overlap being addressed in full. Please ensure that your revision is thorough as failure to address the concerns to our satisfaction may result in your submission not being considered further.

5. We note that you have referenced (unpublished on pages 6 and 27) which has currently not yet been accepted for publication. Please remove this from your References and amend this to state in the body of your manuscript: (ie “Bewick et al. [Unpublished]”) as detailed online in our guide for authors

6. Please clarify the Figure 11 "Fig.11 Forest plot of the pooled estimate of association knowledge about danger sign and 

homebirth preference in Ethiopia, 2022

" in page "21" and Figure 11 "Fig. 11 Forest plot of the pooled estimate of the association between transportation access and homebirth preference in Ethiopia, 2022" in page "22" .

Reviewers' comments:

Reviewer's Responses to Questions

**Comments to the Author**

1. Is the manuscript technically sound, and do the data support the conclusions?

Reviewer #1: Yes

Reviewer #2: Yes

2. Has the statistical analysis been performed appropriately and rigorously? 

Reviewer #1: I Don't Know

Reviewer #2: Yes

3. Have the authors made all data underlying the findings in their manuscript fully available?

Reviewer #1: No

Reviewer #2: No

4. Is the manuscript presented in an intelligible fashion and written in standard English?

Reviewer #1: No

Reviewer #2: Yes

5. Review Comments to the Author

Reviewer #1: Manuscript #: PONE-D-23-02877

Title: PREVALENCE OF HOMEBIRTH PREFERENCE AND ASSOCIATED FACTORSAMONG PREGNANT WOMEN IN ETHIOPIA: SYSTEMATIC REVIEW AND META-ANALYSIS

Article type: Research Article

Comments to authors

The manuscript may improve if the following comments are incorporated very well.

Abstract:

Background:

What is the problem? Is it homebirth itself or not being attended by skilled birth attendant?

Methods:

All parts of method section should be summarized. An important time here is years of publications that included not the time that you conducted searching. Use of funnel plot is subjective.

Conclusion: what is the findings implication?

Background

Authors should indicated contextual differences comparing different geographical, economic and social backgrounds, i.e., global to local.

Why should we care about homebirth? What proportion of morbidity mortality are attributable to home birth?

“Even though Ethiopia has devised Critical strategies including promoting institutional delivery services for lowering maternal morbidity and mortality, just 1 in 4 women gave birth in a medical facility in the country [4].” But the current data in Ethiopia different from this! Update your source of evidence.

“According to the 2019 Mini Ethiopian Demographic and Health Survey report, of all live births in the five years before the survey, only 50% were delivered by a skilled provider [6].” Don’t you think this contradict with above report?

Show the controversies of findings of the studies behind the variation you evidenced/confirmed. In other word, where is/are the variation(s)?

“Therefore, this systematic review and meta-analysis aimed to provide more current and representative national data on homebirth Preference and associated factors in Ethiopia.”

How this provide current data? You pooled already available evidences.

Method

Include PROSPERO registration number.

“The research articles search strategy, selection of studies, data extraction, and result reporting were done in accordance with Preferred Reporting Items for Systematic Reviews and Meta-Analyses (PRISMA) guidelines [11]. A PICO principle was adapted for searching terms. (PRISMA) guidelines were used to state this systematic review and meta-analysis [12].” What the difference between these two references? Which PRISMA guideline was used in your case?

What PICO includes in your case?

Explain databases and non-databases’ search engine separately.

What is the difference between Web of Science and WoS?

How do authors make sure search terms are not missed?

Include sample searching strategy for at least one database. The searching strategies for other databases should be annexed as additional files with its appropriate citation here.

Eligibility:

Clearly indicate the eligibility criteria using bullet or numbering.

“…..included all studies that were conducted on homebirth preference and associated factors among pregnant women in Ethiopia.” what if your outcome and factors are missing?

“The participants were pregnant women/mothers” what do you mean? Did you conduct primary study?

“We included all types of articles that were published in the form of journal articles, master's theses, and dissertations in the English language. Published studies were included in the study, but unpublished studies were not found.” This is eligibility criteria, it is not where you report availability of the unpublished articles. Further, make sure the sentences are clear to audience.

“Full research articles, which were not accessed after at least two email contacts of the primary author, were not included because of the failure to assess the quality of articles in the absence of full text.” I failed to understand the meaning and importance of this sentence.

What is timeframe for included studies? Why studies before 2016 excluded? However, Ethiopia was striving to improve institutional delivery during MDG. The institutional delivery increased from 6% to

“All studies conducted in Ethiopia were included.” This is default.

Outcome

Is the content of this paragraph is only about outcome variables? Rename subtopic.

“The first outcome of interest was the prevalence, which was estimated as the total number of women preferring homebirth, cases divided by the total number of sample sizes multiplied by 100.” Do you mean? How ‘prevalence of home birth preference’ calculated? Did you pool calculated proportion or pool all events and samples to calculate pooled proportion?

Clearly state how you measured all variables (outcome and explanatory).

Data collection and quality assessment

Is ‘data collection’ appropriate here?

What are finding of the Joanna Briggs Institute? How many articles were excluded due to quality? How?

“Three data extractors (JW, EM and ML) extracted data by using a standardized data extraction checklist on Microsoft Excel.” Authors should mention the content of checklist that used for data extraction.

The authors should separately write the data extraction and quality control clearly. Who did the data extraction? And how about the quality of included articles?

“Inconsistencies between three independent reviewers were fixed by including other reviewers (MD, AT, WB and KM) after discussion for possible agreement.” How do think this can happen?

Data analysis and synthesis

“Using a format prepared in a Microsoft Excel spreadsheet, necessary information from each original study was extracted.” You already told us above! Why you repeat same thing here and there. This comments works in many place in your document.

Have conducted sensitivity analysis? How?

“Meta-regression was computed to see the presence of statistically significant heterogeneity.” What is the purpose of meta-regression? When do you conduct it?

Result

Use standard PRISMA flow chart. This is not attractive to audience.

Separately mention articles that obtained from databases and additional sources.

How these three articles can be included if your search is limited with timeframe from the very beginning?

Table 1 How about study setting?

Prevalence of Homebirth Preference:

Why do you think there is high heterogeneity?

“The highest 69.75(95% CI 64.98,74.51) prevalence of homebirth Preference was observed in Zone 3, Afar region [17] and the lowest 24.02(19.87,28.17) prevalence of homebirth Preference was reported in Arba Minch zuria surveillance site, SNNPR [21]” what do you mean? Are these the pooled prevalence?

Figure 2 has small font size and difficult to read.

What aspects of variations should you consider to bring down this high heterogeneity in addition to sample size and publication year?

Publication bias

“The visual examination of the funnel plot presented symmetrically which is an indicator for the absence of publication bias (Fig. 5)” this funnel plot is not symmetry.

Sensitivity analysis

Have you observed the output after excluding Kaysay et al., 2019 article? Authors need to be honest when reporting findings. Tell the audience what you found with and without Kaysay et al., 2019 article.

Factors associated with homebirth preference

In all figures exposure and non-exposure were not appropriately placed. Indicate exposure and non-exposure status in all figures.

Generally the figures need edition to make them attractive to audience. The result sections are full of unnecessary statements and lack focus.

Discussion

The explanations that given to similarities and differences need appropriate and proper citation. The author repeated result in discussion. The discussion lack focus.

Conclusion

Same comment as in abstract.

Reviewer #2: It would be a convincing and smart scientific work if authors are able to address all the given comments. all my comments were given on the PDF files. They are quite readable, understandale and well locayted wehre the issues to be addressed on the mainuscript.

6. PLOS authors have the option to publish the peer review history of their article (what does this mean?). If published, this will include your full peer review and any attached files.

Reviewer #1: **Yes: **DR. Kasiye Shiferaw Gemechu

Reviewer #2: **Yes: **Bayu Begashaw Bekele

---

## [Author Response · Author response to Decision Letter 0]

5 May 2023

Responses to Editor and Reviewers

First of all, we would like to thank the Editor and Reviewers of this manuscript for giving us such constructive comments and questions to enrich this manuscript.

S/N Questions/comments Authors response

 General comments 

1 The language of the manuscript needs to be improved for publication. Language has been revised

2 The problem studied is not well explained in a focused manner? Modified with clear explanation of the problem

3 I am not convinced why only studies published after 2016 are included? I recommend you to include those 3 studies published before 2016. The reason is that the research articles published before 2016 were not representative of the current strategy/policy of the country. This means the reality of the country, the current strategy which is from 2016 is the focused strategy to reduce home birth preference to reduce maternal mortality occurred due to home birth. Thus way the information before this year, 2016 cannot be the representative of the current strategy because the information before the strategy might be occurred due the attention was not given to them to reduce the problem but currently, from 2016 to date full attention is being given and follow up also along the strategy. Thus, the aim of the meta-analysis is to identify the prevalence and factors related to the problem during this focused strategy to prevent homebirth preference which leads to maternal mortality. If these two were mixed, It would be difficult to be representative information with current policy of the country.

4 Please revise the discussion section Revised.

 Reviewer #1 Authors’ response

1 Abstract: 

 Background:

What is the problem? Is it homebirth itself or not being attended by skilled birth attendant? In Ethiopia, homebirth is being attended by unskilled birth attendant; this can be the reason for many complications, maternal and child mortality.

 Method Method part is revised thoroughly

 Conclusion Improving the knowledge of pregnant women about the benefit of health facility delivery and obstetric danger signs is necessary to decrease the prevalence of homebirth preference; for these can reduce negative outcomes occurred during delivery.

2 Background

 Regarding the information raised in background Background is summarized, improved and modified (indicated with track change)

 Methods 

 PROSPERO registration number Still on process

 PRISMA guideline for result writing Citation 11 

 PICO A research question used to conduct this systematic review and meta-analysis.

A research question of this review was developed on PICO principle ( pregnant mothers, factors, condition/setting/ Ethiopia, outcome/home birth)

 Data bases Google Scholar, Medline/Pub Med, Cochrane library, the Web of Science, Hinari, Science Direct, ProQuest, African Journals Online

 Non data bases online university repositories (University of Gondar, Addis Ababa, Jimma and Haramaya University) 

From these online searching (repositories) no researches articles, other than those searched from data bases are found. 

 How do authors make sure search terms are not missed? Before developing search term authors care full read and search for deferent terms with the outcome and factors might be published with the problem under the study

 Eligibility criteria In this systematic review and meta-analysis, we included:-

• All studies that were conducted on homebirth preference and/or associated factors among pregnant women in Ethiopia. 

• all types of articles that were published in the form of journal articles

• Master’s theses and dissertations in the English language.

• The restriction was made for the date of publication, articles published before 2016 were excluded because the Ethiopian strategies to overcome the problem of homebirth were critically started since 2016 as transformation strategies and policy.

 Why studies before 2016 were excluded? The reason is that the research articles published before 2016 were not representative of the current strategy/policy of the country. This means the reality of the country, the current strategy which is from 2016 is the focused strategy to reduce home birth preference to reduce maternal mortality occurred due to home birth. Thus way the information before this year, 2016 cannot be the representative of the current strategy because the information before the strategy might be occurred due the attention was not given to them to reduce the problem but currently, from 2016 to date full attention is being given and follow up also along the strategy. Thus, the aim of the meta-analysis is to identify the prevalence and factors related to the problem during this focused strategy to prevent homebirth preference which leads to maternal mortality. If these two were mixed, It would be difficult to be representative information with current policy of the country.

 Data extraction and quality appraisal Score of quality (table 1) and contents of extraction check lists are explained in manuscripts, study setting is there in table 1 also

 Meta-regression Was done for quantitative variables(year of publication and sample size)

 sensitivity Done to see outliers 

 What aspects of variations should you consider to bring down this high heterogeneity in addition to sample size and publication year?

Publication bias No heterogeneity was detected using meta regression table 2 

 The visual examination of the funnel plot presented symmetrically which is an indicator for the absence of publication bias (Fig. 5)” this funnel plot is not symmetry Egger and begger test shows no publication bias (not significant) as well as the funnel plot is somewhat symmetric even though it is subjective.

 Reviewer #2 Response 

Abstract Study setting Google Scholar, Medline, Pub Med, Cochrane library, and the Web of Science search engines were used to identify research articles for this systematic review and meta-analysis from 20th August 2022 to 6th November 2022, Ethiopia

 Background Modified 

 PICO A research question used to conduct this systematic review and meta-analysis.

A research question of this review was developed on PICO principle ( pregnant mothers, factors, condition/setting/ Ethiopia, outcome/home birth)

 Classify it as inclusion and exclusion criteria. Based on your PICO guideline. In this systematic review and meta-analysis, we included:-

• All studies that were conducted on homebirth preference and/or associated factors among pregnant women in Ethiopia. 

• All types of articles that were published in the form of journal articles

• Master’s theses and dissertations in the English language.

• The restriction was made for the date of publication, articles published before 2016 were excluded because the Ethiopian strategies to overcome the problem of homebirth were critically started since 2016 as transformation strategies and policy.

 What was the reason for not including studies done before 2016? The reason is that the research articles published before 2016 were not representative of the current strategy/policy of the country. This means the reality of the country, the current strategy which is from 2016 is the focused strategy to reduce home birth preference to reduce maternal mortality occurred due to home birth. Thus way the information before this year, 2016 cannot be the representative of the current strategy because the information before the strategy might be occurred due the attention was not given to them to reduce the problem but currently, from 2016 to date full attention is being given and follow up also along the strategy. Thus, the aim of the meta-analysis is to identify the prevalence and factors related to the problem during this focused strategy to prevent homebirth preference which leads to maternal mortality. If these two were mixed, It would be difficult to be representative information with current policy of the country.

---

## [Decision Letter · Decision Letter 1]

17 Jul 2023

PONE-D-23-02877R1PREVALENCE OF HOMEBIRTH PREFERENCE AND ASSOCIATED FACTORSAMONG PREGNANT WOMEN IN ETHIOPIA: SYSTEMATIC REVIEW AND META-ANALYSISPLOS ONE

Dear Dr. Feyisa,

Thank you for submitting your revised manuscript to PLOS ONE. After careful consideration, we feel that it has merit but does not fully meet PLOS ONE’s publication criteria as it currently stands. Therefore, we invite you to submit a revised version of the manuscript that addresses the points raised during the review process. The manuscript has improved a lot considering the given comments and suggestions. However, there are still areas that need revision.

1. The English language of the manuscript need further improvement. It has a lot of vague words that makes difficult to grasp the idea. please try to revise the language with professional assistance from expert unless your manuscript could not meet the language requirement of the journal.

2. The figures are not such attractive, please modify and edit your figures carefully.

3. Please address the reviewer comment carefully.

We look forward to receiving your revised manuscript.

Kind regards,

Biruk Bogale Wolde

Academic Editor

PLOS ONE

Journal Requirements:

Additional Editor Comments:

Dear Authors,

Thank you for taking time and address the comments of editor and reviewers.

Reviewers' comments:

Reviewer's Responses to Questions

**Comments to the Author**

1. If the authors have adequately addressed your comments raised in a previous round of review and you feel that this manuscript is now acceptable for publication, you may indicate that here to bypass the “Comments to the Author” section, enter your conflict of interest statement in the “Confidential to Editor” section, and submit your "Accept" recommendation.

Reviewer #2: All comments have been addressed

Reviewer #3: (No Response)

2. Is the manuscript technically sound, and do the data support the conclusions?

Reviewer #2: Partly

Reviewer #3: Partly

3. Has the statistical analysis been performed appropriately and rigorously? 

Reviewer #2: Yes

Reviewer #3: Yes

4. Have the authors made all data underlying the findings in their manuscript fully available?

Reviewer #2: Yes

Reviewer #3: Yes

5. Is the manuscript presented in an intelligible fashion and written in standard English?

Reviewer #2: Yes

Reviewer #3: Yes

6. Review Comments to the Author

Reviewer #2: My top concern is more about the language issue. It needs further work before sent to acceptance/publication room.

Reviewer #3: Thank you for giving me the opportunity to review the manuscript titled ‘PREVALENCE OF HOMEBIRTH PREFERENCE AND ASSOCIATED FACTORSAMONG PREGNANT WOMEN IN ETHIOPIA: SYSTEMATIC REVIEW AND META-ANALYSIS’

General comment: the manuscript tried to address one of the challenges of maternal and child health in the Ethiopian context and the findings might have some significance for policy implications. By saying this I would try to forward some possible comments and suggestions regarding the manuscript.

• There was another systematic review and meta-analysis on the same topic in 20219 https://www.ncbi.nlm.nih.gov/pmc/articles/PMC8042927/ ) by using 40 studies which had far better strength to estimate the pooled estimate than the current one. So, what are the new findings and methodologies added to the current study?

• I think there were also some issues regarding the proper use of search engines. I think it was possible to use include other studies on facility delivery by inverting their outcome of interest( for example a study that showed a facility delivery of 71% means that a home delivery of 29%). In addition, almost all of the studies were confined to three regions but there were plenty of studies conducted throughout every corner of the country.

• I also doubt the pooled prevalence. A recent and similar study in 2021 showed 66% whereas the current one showed 39.62%. what could be the possible justification behind these discrepancies?

• The majority of the statements throughout the manuscript were vague to understand and need paraphrasing. For instance, in the methodology section ‘Published studies were included in the study, but unpublished studies were not found.’ What does it mean? in another part, it was mentioned that the authors were using grey literature from various repositories???

• The authors failed to mention the search strategy used to retrieve relevant articles from each electronic database. For instance, what were the Medical Subject Heading (MeSH) and keywords used for PubMed, google scholar…

• The result lacks subgroup analysis by region, publication year, and any other significant attributes. This shows that there is a significant homogeneity in the prevalence of home birth across the country which is unlikely and this might be due to an erroneous way of searching and inclusion of the articles.

• The figures were not eye-catching for the readers and try to modify them

• The figures for the exposure variables were a little bit confusing ‘for example the right side for formal education and the left side for no formal education??? It's a new way of data presentation in the forest plot for me. Once you select the odds ratios that deal with formal education, you can indicate it by interpreting the result in the statement form.

• Reporting an excess number of variables as predictors with merely 7 studies by itself shows the spurious type of relations.

• Discussion section

To put a strong public health implication, it would be better to compare the findings from the nationally representative data like EDHS 2016 04 mini EDHS 2019...

• Once, the authors tried to address the above-mentioned basic concerns, the manuscript may have the chance to be published in journal

Thank you!

7. PLOS authors have the option to publish the peer review history of their article (what does this mean?). If published, this will include your full peer review and any attached files.

Reviewer #2: No

Reviewer #3: No

---

## [Author Response · Author response to Decision Letter 1]

26 Jul 2023

Responses to Editor and Reviewers

First of all, we would like to thank the Editor and Reviewers of this manuscript for giving us such constructive comments and questions to enrich this manuscript.

S/N Questions/comments Authors’ response

 General 

 Language and figures Clearly modified

 Reviewer #1 Authors’ response

1 The language of the manuscript needs to be improved for publication. Language has been revised

 Reviewer #3 Authors’ response

1 Other meta-analysis? Home birth and preference of home birth is not the title of similar issue. Homebirth preference is the intention/plan to give birth outside health facilities with the help of unskilled birth attendants.

While home birth is the action which indicates giving birth at home. Intention to give birth at home is not mean giving birth at home but their interest for their coming delivery time but they may or may not give birth at home and it can be one of the factors those causes home birth but not limited to it.

Therefore home birth is not preference of homebirth. 

 language revised

 Searching strategy It was already described clearly in manuscript

 Subgroup analysis Analyzed separately in the manuscript

2 

 Language and pictures revised

---

## [Editor Report · Decision Letter 2]

11 Aug 2023

PONE-D-23-02877R2PREVALENCE OF HOMEBIRTH PREFERENCE AND ASSOCIATED FACTORSAMONG PREGNANT WOMEN IN ETHIOPIA: SYSTEMATIC REVIEW AND META-ANALYSISPLOS ONE

Dear Dr. Feyisa,

Thank you for submitting your manuscript to PLOS ONE. After careful consideration, we feel that it has merit but does not fully meet PLOS ONE’s publication criteria as it currently stands. Therefore, we invite you to submit a revised version of the manuscript that addresses the points raised during the review process.

ACADEMIC EDITOR: Dear Authors, Thank you for submitting your revised version of the manuscript. The manuscript has improved a lot during this revision. However, there are remaining issues to be addressed before proceeding to the next step. Please address the following comments:Title:Comment: Please do not use upper case letters for all words in the title. Use the upper-case letter for the first letter of the word. Abstract:
Methods: There is a sentence “From 20th August 2022 to 6th November 2022, Google Scholar, Medline, PubMed, Cochrane Library, and Web of Science were used to find research articles for this systematic review and meta-analysis.” Please paraphrase as “Search of Google Scholar, Medline, PubMed, Cochrane Library and Web of Science were done for this study from 20th August 2022 to 6th November 2022.”
Please paraphrase “To find out heterogeneity, Cochrane Q test statistics and I2 statistics were used.”  Sentence as “Cochrane Q test statistics and I2 statistics were used to check heterogeneity of the studies.”
Conclusion:
Please narrate the public health implication of the pooled prevalence of 39.62% in one sentence.Methods:Search StrategyThere is a statement “……as well as it was sent for registration, and ID 422354 was assigned.” If the protocol is registered mention it and if not remove this statement. Eligibility Criteria
Please remove bullets from the listed criteria and use Arabic numerals.
There is a statement read “The restriction was made for the date of publication. Articles published before 2016 were excluded because the Ethiopian strategies to overcome the problem of homebirth were critically started in 2016 as transformation strategies and policies.” Please remove this statement since it has factual and logical errors. Ethiopia had been doing a wonderful job to decrease home birth even before 2016 through HEP and expanding healthcare facilities. Results:
Please paraphrase “976 studies in all were found using a variety of electronic sources and library catalogs. 646 articles recorded from these studies were found to be duplicates and were eliminated. 303 irrelevant research publications were excluded from our analysis after being reviewed for titles and abstracts.” this statement. �
Figure 1: Check the title of the figure date? It is 2023 now. �
Table 1: Check the title of the Table. It should be self-explanatory (what, where and when).
Under “Association between ANC follow up and Homebirth preference” subtitle line 1, please remove the full stop after the citation. Discussion:
The pooled prevalence of this SRMA is compared to the individual or pocket study conducted in Nigeria and Tanzania. How a SRMA is compared with pocket study? This should be compared with another SRMA!! Please revise your discussion accordingly. Conclusion:
Please narrate the public health implications of the pooled prevalence of homebirth intention in one sentence. 

Please submit your revised manuscript with track changed word format by Sep 25 2023 11:59PM. If you will need more time than this to complete your revisions, please reply to this message or contact the journal office at plosone@plos.org. Please include the following items when submitting your revised manuscript:A rebuttal letter that responds to each point raised by the academic editor and reviewer(s). You should upload this letter as a separate file labeled 'Response to Reviewers'.A marked-up copy of your manuscript that highlights changes made to the original version. You should upload this as a separate file labeled 'Revised Manuscript with Track Changes'.An unmarked version of your revised paper without tracked changes. You should upload this as a separate file labeled 'Manuscript'.If applicable, we recommend that you deposit your laboratory protocols in protocols.io to enhance the reproducibility of your results. Protocols.io assigns your protocol its own identifier (DOI) so that it can be cited independently in the future. For instructions see: https://journals.plos.org/plosone/s/submission-guidelines#loc-laboratory-protocols. Additionally, PLOS ONE offers an option for publishing peer-reviewed Lab Protocol articles, which describe protocols hosted on protocols.io. Read more information on sharing protocols at https://plos.org/protocols?utm_medium=editorial-email&utm_source=authorletters&utm_campaign=protocols.

We look forward to receiving your revised manuscript.

Kind regards,

Biruk Bogale Wolde

Academic Editor

PLOS ONE

Journal Requirements:

Additional Editor Comments:

Dear Authors,

Thank you for submitting the revised version of your manuscript. The manuscript has improved a lot during this revision. However, there are still issues to be revised before proceeding further steps.

Please address the following comments:

Title:

Comment: Please do not use upper case letters for all words in the title. Use the upper-case letter for the first letter of the word.

Abstract:

Methods: There is a sentence “From 20th August 2022 to 6th November 2022, Google Scholar, Medline, PubMed, Cochrane Library, and Web of Science were used to find research articles for this systematic review and meta-analysis.” Please paraphrase as “Search of Google Scholar, Medline, PubMed, Cochrane Library and Web of Science were done for this study from 20th August 2022 to 6th November 2022.”

Please paraphrase “To find out heterogeneity, Cochrane Q test statistics and I2 statistics were used.” Sentence as “Cochrane Q test statistics and I2 statistics were used to check heterogeneity of the studies.”

Conclusion:

Please narrate the public health implication of the pooled prevalence of 39.62% in one sentence.

Methods:

Search Strategy

There is a statement “……as well as it was sent for registration, and ID 422354 was assigned.” If the protocol is registered mention it and if not remove this statement.

Eligibility Criteria

Please remove bullets from the listed criteria and use Arabic numerals.

There is a statement read “The restriction was made for the date of publication. Articles published before 2016 were excluded because the Ethiopian strategies to overcome the problem of homebirth were critically started in 2016 as transformation strategies and policies.” Please remove this statement since it has factual and logical errors. Ethiopia had been doing a wonderful job to decrease home birth even before 2016 through HEP and expanding healthcare facilities.

Results:

Please paraphrase “976 studies in all were found using a variety of electronic sources and library catalogs. 646 articles recorded from these studies were found to be duplicates and were eliminated. 303 irrelevant research publications were excluded from our analysis after being reviewed for titles and abstracts.” this statement.

Figure 1: Check the title of the figure date? It is 2023 now.

Table 1: Check the title of the Table. It should be self-explanatory (what, where and when).

Under “Association between ANC follow up and Homebirth preference” subtitle line 1, please remove the full stop after the citation.

Discussion:

The pooled prevalence of this SRMA is compared to the individual or pocket study conducted in Nigeria and Tanzania. How a SRMA is compared with pocket study? This should be compared with another SRMA!! Please revise your discussion accordingly.

Conclusion:

Please narrate the public health implications of the pooled prevalence of homebirth intention in one sentence.

---

## [Author Response · Author response to Decision Letter 2]

14 Aug 2023

Responses to Editor 

First of all, we would like to thank the Editor of this manuscript for giving us such constructive comments and questions to enrich this manuscript.

S/N Questions/comments Authors’ response

 comments 

1 Title. Revised

 abstract Revised 

1 Methods Revised indicated by track change within manuscript

 Result and conclusion Revised indicated by track change within manuscript

---

## [Editor Report · Decision Letter 3]

29 Aug 2023

Prevalence of Homebirth Preference and Associated Factors among Pregnant Women in Ethiopia: Systematic Review and Meta-Analysis

PONE-D-23-02877R3

Dear Dr. Jira Wakoya,

We’re pleased to inform you that your manuscript has been judged scientifically suitable for publication and will be formally accepted for publication once it meets all outstanding technical requirements.

Kind regards,

Biruk Bogale Wolde

Academic Editor

PLOS ONE
---

## [Editor Report · Acceptance letter]

12 Sep 2023

PONE-D-23-02877R3 

Prevalence of Homebirth Preference and Associated Factors among Pregnant Women in Ethiopia: Systematic Review and Meta-Analysis 

Dear Dr. Feyisa:

I'm pleased to inform you that your manuscript has been deemed suitable for publication in PLOS ONE. Congratulations! Your manuscript is now with our production department. 

Kind regards, 

on behalf of

Mr Biruk Bogale Wolde 

Academic Editor

PLOS ONE